# Gasdermin-A3 pore formation propagates along variable pathways

Stefania A. Mari [1,3], Kristyna Pluhackova [1,3✉], Joka Pipercevic[2], Matthew Leipner [1], Sebastian Hiller [2], Andreas Engel [1] & Daniel J. Müller [1✉]

Gasdermins are main effectors of pyroptosis, an inflammatory form of cell death. Released by proteolysis, the N-terminal gasdermin domain assembles large oligomers to punch lytic pores into the cell membrane. While the endpoint of this reaction, the fully formed pore, has been well characterized, the assembly and pore-forming mechanisms remain largely unknown. To resolve these mechanisms, we characterize mouse gasdermin-A3 by high-resolution time-lapse atomic force microscopy. We find that gasdermin-A3 oligomers assemble on the membrane surface where they remain attached and mobile. Once inserted into the membrane gasdermin-A3 grows variable oligomeric stoichiometries and shapes, each able to open transmembrane pores. Molecular dynamics simulations resolve how the membrane-inserted amphiphilic β-hairpins and the structurally adapting hydrophilic head domains stabilize variable oligomeric conformations and open the pore. The results show that without a vertical collapse gasdermin pore formation propagates along a set of multiple parallel but connected reaction pathways to ensure a robust cellular response.

[1] Department of Biosystems Science and Engineering, Eidgenössische Technische Hochschule (ETH) Zurich, 4058 Basel, Switzerland. [2] Biozentrum, University of Basel, 4056 Basel, Switzerland. [3] These authors contributed equally: Stefania A. Mari, Kristyna Pluhackova. ✉email: kristyna.pluhackova@bsse.ethz.ch; daniel.mueller@bsse.ethz.ch

Pyroptosis, an inflammatory type of programmed cell death, is characterized by cell swelling, cell membrane disruption, and release of cytoplasmic content, which includes pro-inflammatory molecules, such as the mature forms of interleukin-1β and -18[1–4]. Until recently, it has been thought that pyroptosis is solely linked to caspases, which are activated by inflammasomes, multi-protein complexes that assemble upon recognizing certain danger- or pathogen-associated molecular patterns by cytosolic receptors[5,6]. While the basis of inflammasome assembly and caspase activation has been well established, the exact mechanism of pyroptosis remained unclear for over a decade. In 2015, studies identified gasdermin-D (GSDMD) as an essential mediator of pyroptosis in human and murine cells[1,2,7]. These studies followed by others have shown that the members of the gasdermin family, including GSDMA, GSDMB, GSDMC, GSDMD, and GSDME (or DFNA5)[8–12], are the sole executor of pyroptosis[3,6]. Gasdermin induced pyroptosis plays prominent roles in various hereditary diseases, inflammatory disorders, cancer[6,13,14] and, as recently shown, also in coronavirus-induced diseases[15,16]. Furthermore, gasdermins might function as tumor suppressors by complementing the mitochondrial apoptotic pathway[10] and enhancing cytotoxic lymphocyte responses[17,18].

Most gasdermins share a two-domain architecture of a cytotoxic N-terminal domain (GSDM[Nterm]), which exposes intrinsic membrane pore-forming activity, and an inhibitory C-terminal domain (GSDM[Cterm]), which acts as GSDM[Nterm] repressor. In the inactive state of gasdermin, both domains are connected by a polypeptide linker[3,6]. Canonical and non-canonical inflammasome pathways sense pathogen-derived or host-derived danger signals and activate proteases, mostly caspases, to cleave gasdermin, which releases cytotoxic GSDM[Nterm] from intramolecular autoinhibition by GSDM[Cterm] to induce pyroptosis[2,5,7]. GSDM[Nterm] from different family members exhibit distinct lipid-binding properties to target specific cellular membranes[8,9,11,19,20]. The crystal structure of full-length murine GSDMA3 (mGSDMA3) monomers[8] and the cryo-transmission electron microscopy (cryo-TEM) structure of solubilized ring-shaped mGSDMA3[Nterm] oligomers[21] show the head domain of mGSDMA3[Nterm] to remain largely unaltered, whereas the formation of the two long β-hairpins that outline the transmembrane β-barrel is associated with marked conformational changes. Cryo-TEM also shows that ring-shaped mGSDMA3[Nterm] oligomers have 26- to 28-fold symmetry in contrast to human GSDMD[Nterm] (hGSDMD[Nterm]) oligomers showing 31- to 34-fold symmetry[12,21]. Interestingly, the cryo-TEM of solubilized mGSDMA3[Nterm] and hGSDMD[Nterm] reveals ring-shaped oligomers with and without β-barrel. While ring-shaped oligomers with β-barrel are considered to represent the membrane-inserted state, oligomers without β-barrel are speculated to represent a soluble state before membrane insertion[12,21]. Moreover, negative-stain TEM[8,22] and cryo-TEM[9] of hGSDMD[Nterm] in liposomes show hGSDMD[Nterm] to assemble oligomers of different shapes and sizes. This observation has been confirmed by atomic force microscopy (AFM), which shows hGSDMD[Nterm] forming arc-, slit-, and ring-shaped oligomers in supported lipid membranes (SLM)[9,22]. The ratio of arc-, slit-, and ring-shaped hGSDMD[Nterm] oligomers depends on the lipid composition of the membrane and possibly also on the incubation conditions[22]. However, so far it has not been shown whether the ability of hGSDMD[Nterm] to assemble oligomeric pores of different sizes and shapes also applies to other gasdermin family members. Moreover, it remains unclear which properties of GSDM[Nterm] enable the assembly of such oligomeric variety. Finally, how GSDM[Nterm] oligomers assemble and insert in the membrane and how they transit from the plugged pre-pore state to the lytic, open pore state is not known.

As hGSDMD[Nterm] can assemble various oligomeric shapes in membranes, each able to open transmembrane pores, we here want to understand whether this flexibility in assembly and pore formation can be generalized to other gasdermin family members and, importantly, on which molecular mechanisms the cytolytic oligomeric assembly and pore formation are based. We thus characterize the oligomeric assembly and pore formation of mGSDMA3 in lipid membranes by high-resolution time-lapse AFM and by TEM. The observed variable oligomeric sizes, shapes, and states are analyzed by all-atom and coarse-grained molecular dynamics (MD) simulations to mechanistically understand how mGSDMA3[Nterm] oligomers attach to the membrane, upon insertion into the membrane reassemble and grow, and transit from the plugged pre-pore to the open pore state.

## Results

**mGSDMA3[Nterm] membrane insertion and oligomerization.** So far, the protease that cleaves mGSDMA3 into mGSDMA3[Nterm] and mGSDMA3[Cterm] is not known[6,21], although cleavage of engineered mGSDMA3 can trigger pyroptosis[7]. Hence, to proteolytically cleave mGSDMA3, we inserted a tobacco etch virus protease (TEV) cleavage site into the polypeptide loop connecting the N- and C-terminal domains (Methods, Supplementary Fig. 1a, b). To characterize the insertion and assembly of mGSDMA3[Nterm] by high-resolution AFM, we prepared liposomes from *Escherichia coli* polar lipid extract because mGSDMA3[Nterm] exhibits severe toxicity to bacteria including *E. coli*[8] and gasdermins disrupt mitochondrial membranes showing similar lipid compositions[23,24]. The liposomes were adsorbed onto freshly cleaved atomically flat muscovite mica in adsorption buffer solution, where they collapsed and fused into defect-free supported lipid membranes (SLM; Supplementary Fig. 1c–e). We then incubated the SLMs for 3 h at 37 °C with imaging buffer solution containing 1.5 μM mGSDMA3, which had been beforehand cleaved with 0.4 μM TEV overnight at 37 °C (Supplementary Fig. 1b, f). After incubation, the sample was rinsed with protein-free imaging buffer solution and imaged by either contact mode AFM[25] or force-distance curve-based AFM (FD-based AFM)[26,27] in imaging buffer solution at room temperature (Fig. 1, Supplementary Fig. 2a, b).

The AFM topographs showed arc-, slit-, and ring-shaped mGSDMA3[Nterm] oligomers inserted into SLMs (Fig. 1a–e). Each of the oligomers could either form pre-pores that were plugged or open pores that penetrated the lipid membrane by ≥ 2.0 nm (Fig. 1f, g). The diameter of the ring-shaped oligomers distributed widely from ≈ 20 to 34 nm with a mean of 26.3 ± 2.2 nm (mean ± SD; $n = 149$) (Fig. 1h). Similarly, the diameters of the arc- and slit-shaped oligomers varied considerably in size. Arc-shaped oligomers frequently co-assembled with each other forming larger clusters. Alternatively, two arc-shaped oligomers 'touched' at their ends forming a slit-shaped oligomer.

The different oligomeric shapes, sizes and clusters demonstrate a high structural flexibility of mGSDMA3[Nterm] in forming supramolecular assemblies. Interestingly, the height analysis shows that all types of mGSDMA3[Nterm] oligomers forming pre-pores protruded by the same distance from the membrane surface: 3.4 ± 0.2 nm (mean ± SD; $n = 147$) for arcs, 3.5 ± 0.3 nm ($n = 127$) for slits, and 3.4 ± 0.3 nm ($n = 139$) for rings (Fig. 1i, Supplementary Table 1). Also, mGSDMA3[Nterm] oligomers forming open, lytic transmembrane pores protruded by the same height from the membrane, thus suggesting that the oligomers transit from the plugged pre-pore to the open pore state in the absence of a vertical collapse. Control experiments incubating SLMs with either non-cleaved (full-length) mGSDMA3 or TEV

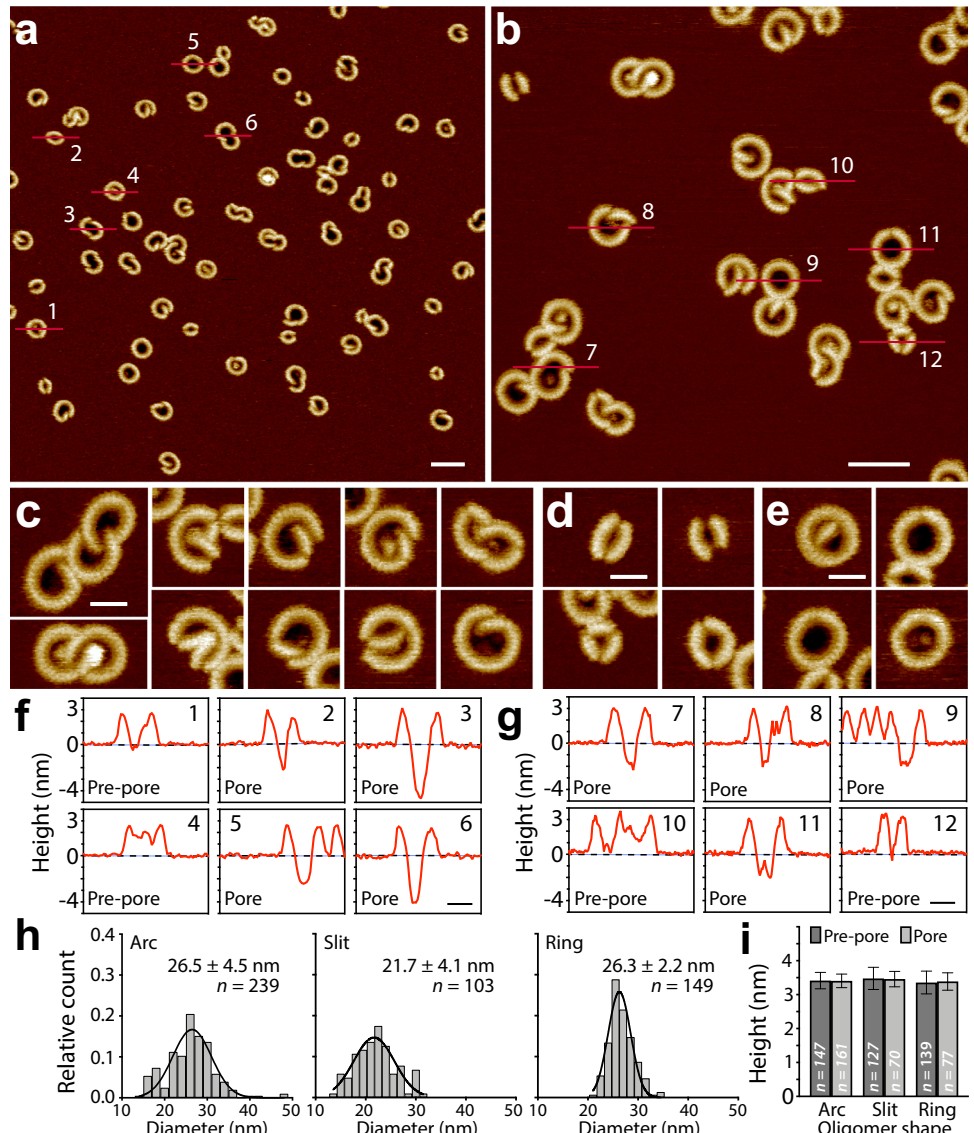

**Fig. 1 Murine GSDMA3[Nterm] assembles a variety of oligomeric shapes and sizes, which can adopt either the pre-pore or pore state, in lipid membranes. a**, **b** AFM topographs of mGSDMA3[Nterm] oligomers formed on supported lipid membranes (SLMs) made from *E. coli* polar lipid extract. **a–e**, AFM topographs showing (**c**) arc-, (**d**) slit-, and (**e**) ring-shaped mGSDMA3[Nterm] oligomers. **f**, **g** Height profiles of mGSDMA3[Nterm] oligomers measured along the red lines indicated in the AFM topographs (**a**, **b**). Numbers in the upper right corner correlate height profiles to red lines in topographs. Black dashed lines represent the membrane surface (0 nm height). Pre-pore and pore states are indicated for each oligomer. Pore state has been defined for oligomers, whose pores protrude $\geq$ 2 nm into the lipid membrane. The FD-based AFM topographs were recorded in imaging buffer solution and exhibit a full-range color scale corresponding to a vertical scale of 6 nm. Scale bars, 50 nm (**a**, **b**) and 20 nm (**c–g**). **h**, Diameters of maximum height of arc-, slit- and ring-shaped mGSDMA3[Nterm] oligomers imaged by AFM in membranes made from *E. coli* polar lipid extract. Black curves represent Gaussian fits determining the mean ± SD values given. **i** Maximum heights of arc-, slit- and ring-shaped mGSDMA3[Nterm] oligomers protruding from the lipid membrane surface and either residing in the pre-pore or pore state. Values present mean ± SD and are given in Supplementary Table 1. *n* gives the number of oligomers analyzed.

alone, did not show oligomers (Supplementary Fig. 2c, d), confirming that they assemble from mGSDMA3[Nterm].

**Membrane-attached and membrane-inserted mGSDMA3[Nterm] oligomers**. Next, we monitored the assembly of mGSDMA3[Nterm] oligomers on SLMs by time-lapse AFM (Fig. 2, Supplementary Fig. 3 and Movie 1). To this end, we injected a solution of 1.5 μM mGSDMA3, which had been beforehand cleaved with 0.4 μM TEV overnight at 37 °C, on a SLM through the AFM fluid chamber. We then imaged the SLM in the imaging buffer solution containing the cleaved mGSDMA3[Nterm] by time-lapse FD-based

AFM[26,27] at 37 °C. We observed arc-, slit- and ring-shaped oligomers assembling and inserting into the membrane. The mGSDMA3[Nterm] oligomers remained relatively immobile and largely kept their position over the time course of the experiment while growing larger oligomers and clusters. The insertion and assembly of differently shaped and sized mGSDMA3[Nterm] oligomers continued until they densely covered the membrane. After 2–3 h of incubation, as the oligomers increased density on the SLM, we observed mobile ring-shaped oligomers, which suddenly could appear, change position or disappear. Interestingly, the mobile class of ring-shaped oligomers was not observed to penetrate the membrane to form open transmembrane pores

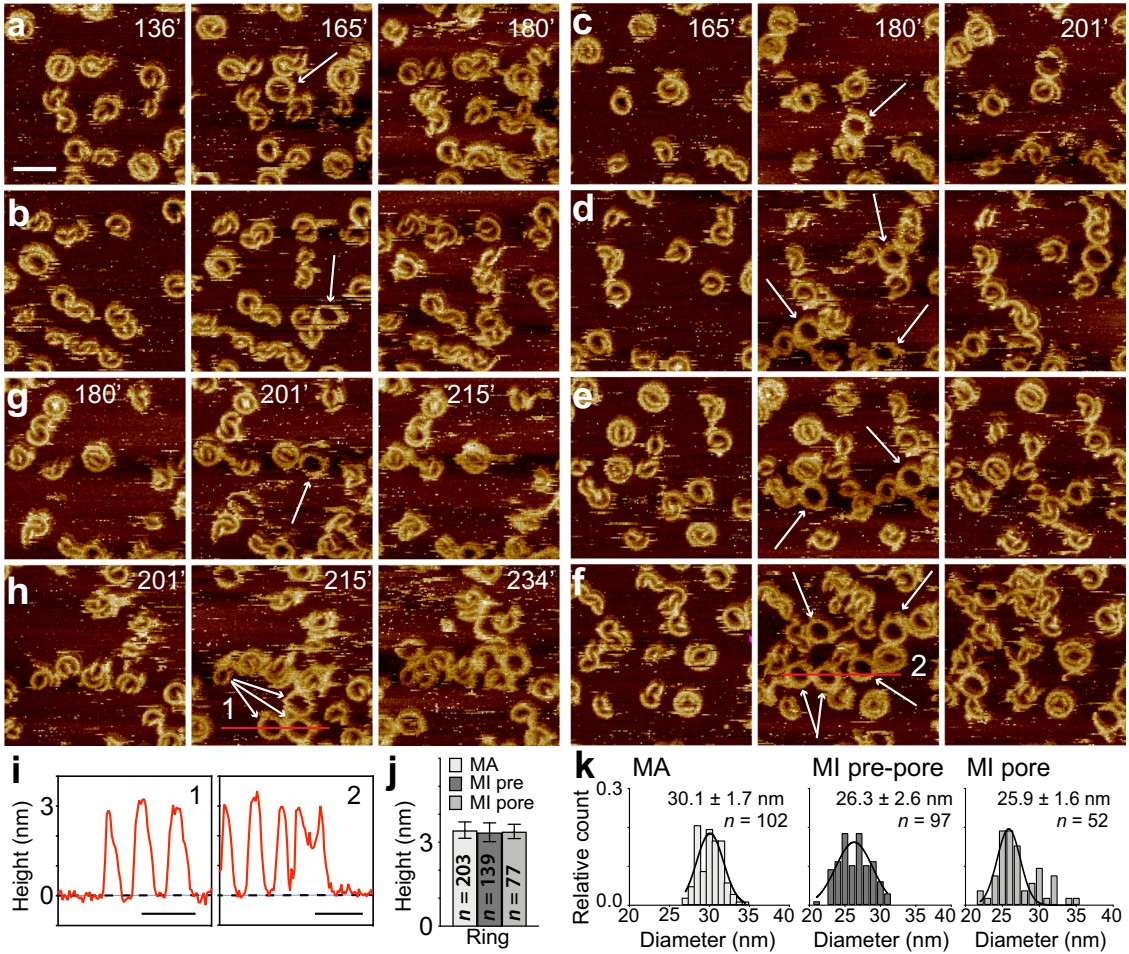

**Fig. 2 Time-lapse AFM sequences showing mobile membrane-attached and immobile membrane-inserted mGSDMA3^Nterm oligomers. a–h** A defect-free SLM made from *E. coli* polar lipid extract was incubated with imaging buffer solution containing 1.5 µM mGSDMA3, which had been beforehand cleaved with 0.4 µM TEV overnight at 37 °C, and imaged in the same solution at 37 °C. Recorded at different time points of the incubation (time stamps indicate minutes), the time-lapse AFM topographs monitor the assembly, disassembly and diffusion of mGSDMA3^Nterm oligomers. From **a** to **h**, the topographs follow over the time course different areas of the SLM. The central topographs capture mobile ring-shaped mGSDMA3^Nterm oligomers (indicated by white arrows) which were not there previously (topograph on the left side) and which thereafter disassemble or change position (topograph on the right side). The time-lapse FD-based AFM topographs were recorded in imaging buffer solution at 37 °C ("Methods"), and their full-range color scale corresponds to a vertical scale of 6 nm. Scale bar of 50 nm applies to all topographs. **i,** Height profiles of mobile mGSDMA3^Nterm oligomers measured along the red lines indicated in the AFM topographs (**f**, **h**). Numbers in the upper right corner correlate height profiles to red lines in topographs. Black dashed lines represent the membrane surface (0 nm height). **j** Maximum heights of ring-shaped oligomers residing in the membrane-attached (MA) and membrane-inserted pre-pore (MI pre) and pore (MI pore) state. Values present mean ± SD and are given in Supplementary Table 1. **k** Diameters of the maximum height of ring-shaped mGSDMA3^Nterm oligomers residing in the membrane-attached (MA) and membrane-inserted pre-pore (MI pre-pore) and pore (MI pore) state and imaged by time-lapse AFM. Black curves represent Gaussian fits determining the mean ± SD values given. *n* gives the number of oligomers analyzed. For further information on the representative time-lapse AFM series see Supplementary Fig. 3 and Movie 1.

(Fig. 2i). The mobile oligomers and the immobile oligomers residing in the pre-pore and pore state protruded by the same height above the membrane (Fig. 2j, Supplementary Table 1). The fact that mobile and immobile oligomers residing in the pre-pore and pore state protruded by the same height from the membrane suggests that membrane attachment, insertion, assembly, growth, and lytic pore formation occur in the absence of a vertical collapse.

We then rinsed the SLM with protein-free imaging buffer solution. As a result, the mobile ring-shaped mGSDMA3^Nterm oligomers disappeared thus demonstrating that they were weakly attached to and not inserted into the lipid membrane (Supplementary Fig. 4). The relatively immobile arc-, slit-, and ring-shaped oligomers remained after rinsing, which confirms that they are inserted into the membrane and reside either in the plugged pre-pore or the open pore state (Fig. 1). We found these membrane-inserted mGSDMA3^Nterm oligomers to be stable for over nine days (Supplementary Fig. 5), which indicates that the various oligomeric shapes and sizes co-exist in equilibrium after membrane insertion. The mobile ring-shaped oligomers, which can be washed away, resemble the previously introduced cryo-TEM structures of mGSDMA3^Nterm and hGSDMD^Nterm ring-shaped oligomers without transmembrane β-barrel[12,21]. Our AFM topographs also show that mobile ring-shaped oligomers have slightly larger diameters compared to immobile ring-shaped oligomers (Fig. 2k). This finding also agrees with the cryo-TEM data reporting that ring-shaped oligomers with β-barrel have a smaller diameter compared to ring-shaped oligomers without β-barrel[12]. We hence speculate that the mobile membrane-attached oligomers observed by AFM represent the solubilized mGSDMA3^Nterm oligomers without transmembrane β-barrel observed by cryo-TEM[21].

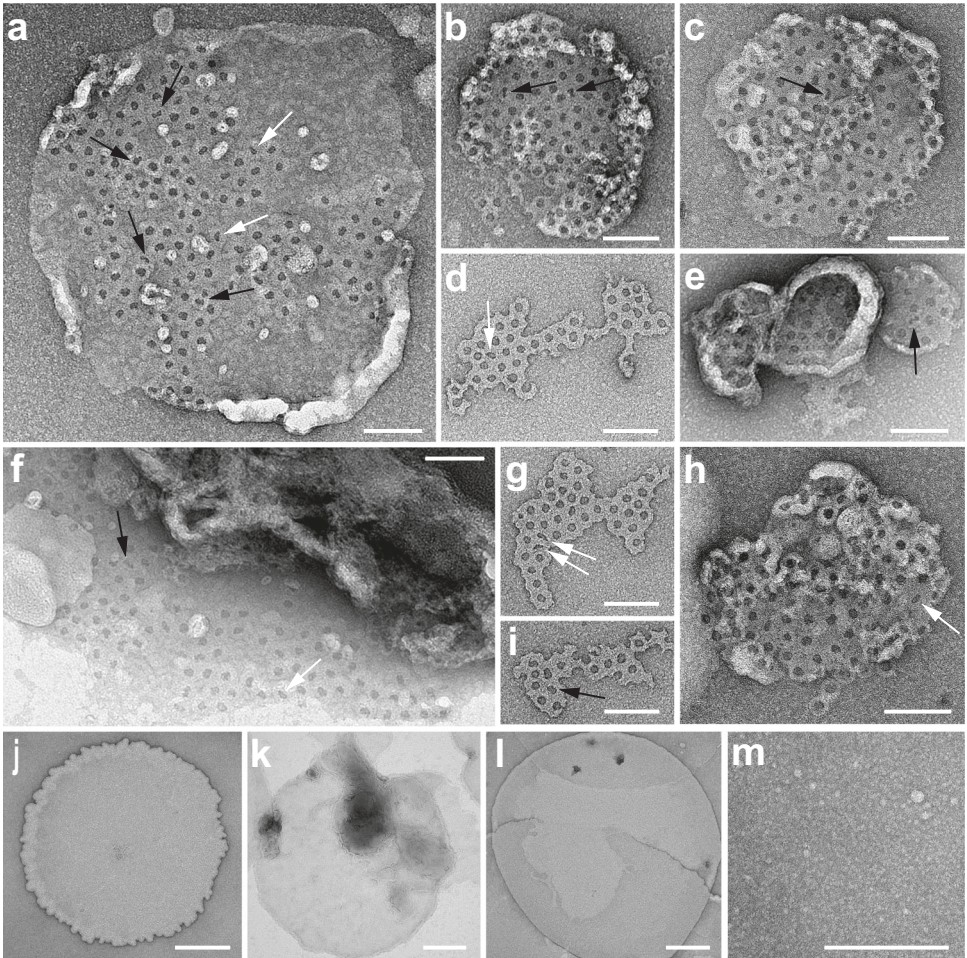

**Fig. 3 Transmission electron microscopy (TEM) of negative stained mGSDMA3[Nterm] oligomers inserted into liposomes made from *E. coli* polar lipid extract. a–i** TEM images of liposomes after incubation with mGSDMA3 in the presence of TEV. Upon adsorption onto TEM grids the liposomes fused into membrane patches that spread out over the carbon film. Most mGSDMA3[Nterm] oligomers inserted into the lipid membranes appear ring-shaped but arc-shaped (black arrows) and slits-shaped (white arrows) oligomers are observed as well. Suspended liposomes made from *E. coli* polar lipid extract were incubated overnight at 37 °C with 7 µM mGSDMA3 and 1.5 µM TEV. **j–l** TEM images showing large vesicles fused on the carbon film from empty *E. coli* polar lipid liposomes (**j**), from liposomes incubated with mGSDMA3 (7 µM) (**k**), and from liposomes incubated with TEV (1.5 µM) (**l**). **m** The mixture of mGSDMA3 and TEV incubated in the absence of liposomes exhibits a granular appearance. None of the samples (**j–m**) showed arc-, slit-, or ring-shaped oligomers as observed for mGSDMA3[Nterm] incubated with liposomes. All samples (**a–m**) were incubated overnight at 37 °C, adsorbed to glow-discharged carbon coated grids, negatively stained with uranyl acetate and imaged at 120 kV (Methods). Scale bars, 100 nm (**a–i**) and 200 nm (**j–m**).

**Oligomerization in liposomes**. To test whether mGSDMA3[Nterm] also assembles different oligomeric shapes and sizes in free liposomes, we incubated lipid vesicles from *E. coli* polar lipid extract with mGSDMA3[Nterm] and recorded TEM images of negatively stained samples (Fig. 3a–i). The images confirmed the assembly of oligomers of different shapes and sizes. The negative stain penetrating into the different oligomeric shapes indicates that the oligomers formed lytic transmembrane pores. The TEM controls also showed that neither non-cleaved mGSDMA3 in the presence of lipid membranes nor that mGSDMA3[Nterm] in the absence of lipid membranes assembles arc-, slit-, and ring-shaped oligomers (Fig. 3j–m), thus confirming the necessity of mGSDMA3[Nterm] and lipid membrane for the assembly of oligomers.

**Plasticity and pore formation of mGSDMA3[Nterm] oligomers**. Intrigued by the ability of mGSDMA3[Nterm] to form oligomers of different shapes and sizes, we directly visualized the membrane-inserted oligomers by high-resolution AFM (Fig. 4a). We then determined the number of subunits of each ring-shaped oligomer

and calculated correlation averages of oligomers having the same stoichiometry (Fig. 4a–d). The analysis revealed stoichiometries ranging from 18 to 36, with the most frequent stoichiometry of 30 (Fig. 4b). This rather wide distribution of stoichiometries, which considerably expands the 26- to 28-fold stoichiometry reported earlier for ring-shaped mGSDMA3[Nterm] oligomers[21], may have originated from different sample preparations and lipid compositions, as both parameters can influence the self-assembly of gasdermins[21,22]. The number of mGSDMA3[Nterm] assembling ring-shaped oligomers scaled non-linearly with the outer ring diameter (Fig. 4b). Also, the outer distance between the mGSDMA3[Nterm] forming ring-shaped mGSDMA3[Nterm] oligomers changed with the stoichiometry (Fig. 4c). Whereas the outer distance between mGSDMA3[Nterm] taken from ring-shaped oligomers showing 28-fold stoichiometry agreed well with the distance between mGSDMA3[Nterm] in the atomistic model of oligomeric rings showing 27-fold and 28-fold stoichiometries[21], the distance between mGSDMA3[Nterm] from much smaller and larger ring-shaped oligomers did not. These results thus highlight a considerable structural plasticity of mGSDMA3[Nterm] to assemble oligomers of various shapes and sizes.

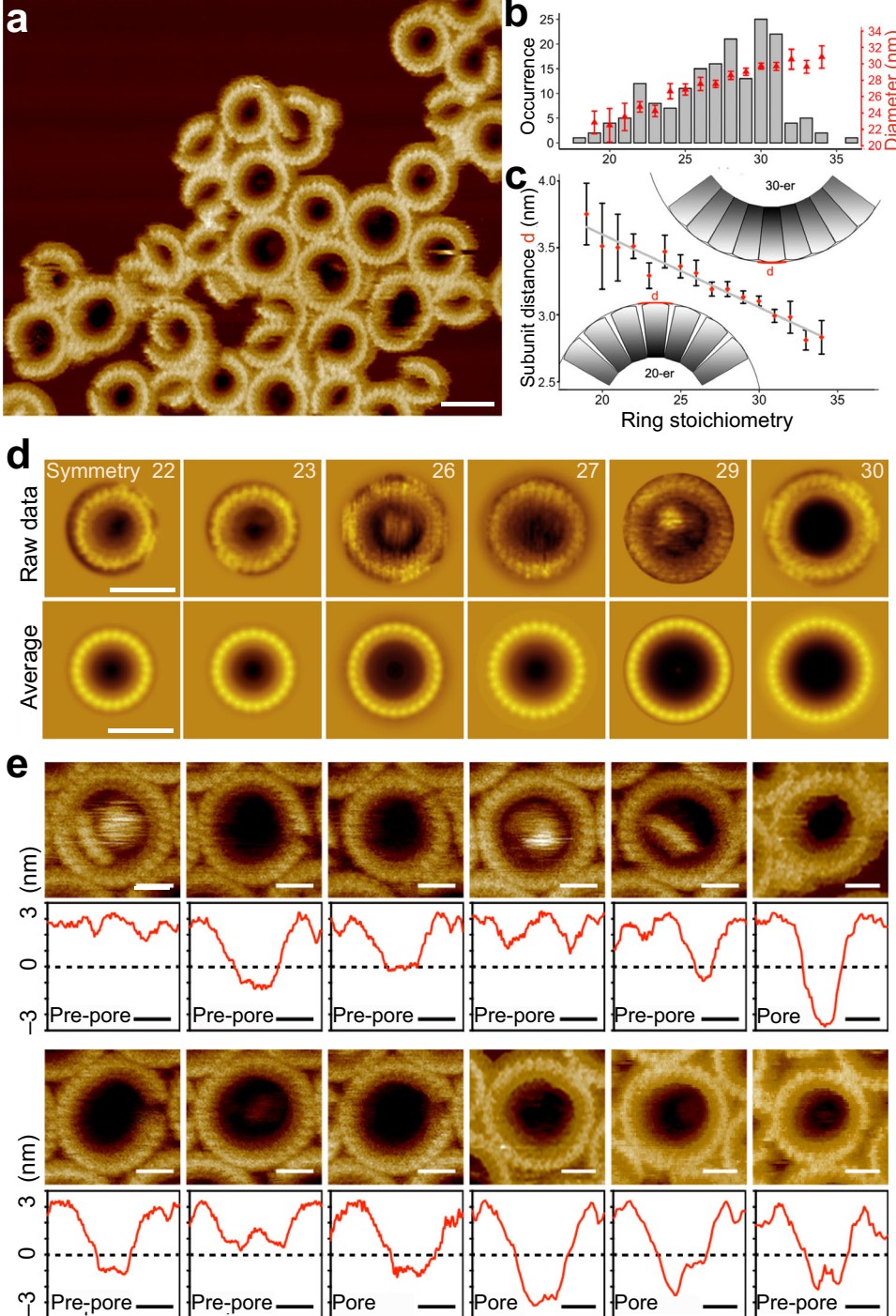

**Fig. 4 mGSDMA3$^{Nterm}$ features high structural flexibility to assemble ring-shaped oligomers of different stoichiometries that show different stages of pore formation. a** High-resolution FD-based AFM topograph of mGSDMA3$^{Nterm}$ oligomers inserted in SLMs made from *E. coli* polar lipid extract. Visible are individual mGSDMA3$^{Nterm}$ assembling variably sized and shaped oligomers. **b** Left axis, occurrence of ring-shaped mGSDMA3$^{Nterm}$ oligomers ($n = 174$) showing symmetries ranging from 18 to 36. The most prominent stoichiometry observed is 30, followed by 31, 28, 27, 26 and 29. Right axis, outer diameter of ring-shaped mGSDMA3$^{Nterm}$ oligomers having different stoichiometries. **c** Distance between mGSDMA3$^{Nterm}$ at the outer periphery of ring-shaped oligomers having different stoichiometries. Distances were estimated from the outer diameter of ring-shaped oligomers (**b**) divided by the number of mGSDMA3$^{Nterm}$. Data points (red) in (**b, c**) give averages and errors SE of 174 ring-shaped mGSDMA3$^{Nterm}$ oligomers analyzed. Insets show a hypothetical packing of mGSDMA3$^{Nterm}$ arranged into ring-shaped oligomers having 20 and 30 stoichiometries. **d** Correlation averaged AFM topographs. Top row, selected high-resolution contact mode and FD-based AFM topographs of individual ring-shaped mGSDMA3$^{Nterm}$ oligomers inserted into SLMs and having different stoichiometries ranging from 22 to 30. Bottom row, symmetrized correlation averages of the exemplified AFM topographs showing stoichiometries ranging from 22 to 30 (Methods). **e** High-resolution contact mode and FD-based AFM topographs and height profiles of individual ring-shaped mGSDMA3$^{Nterm}$ oligomers inserted in lipid membranes. Oligomers residing in the pre-pore and pore state protrude at same height above the lipid membrane (see Fig. 1). Red lines indicate height profiles, black dashed lines indicate the SLM surface (0 nm height). The full-range color scale of the topographs corresponds to a vertical scale of 8 nm (**a**) and 6 nm (**d, e**). Scale bars, 30 nm (**a**), 20 nm (**d**), and 10 nm (**e**).

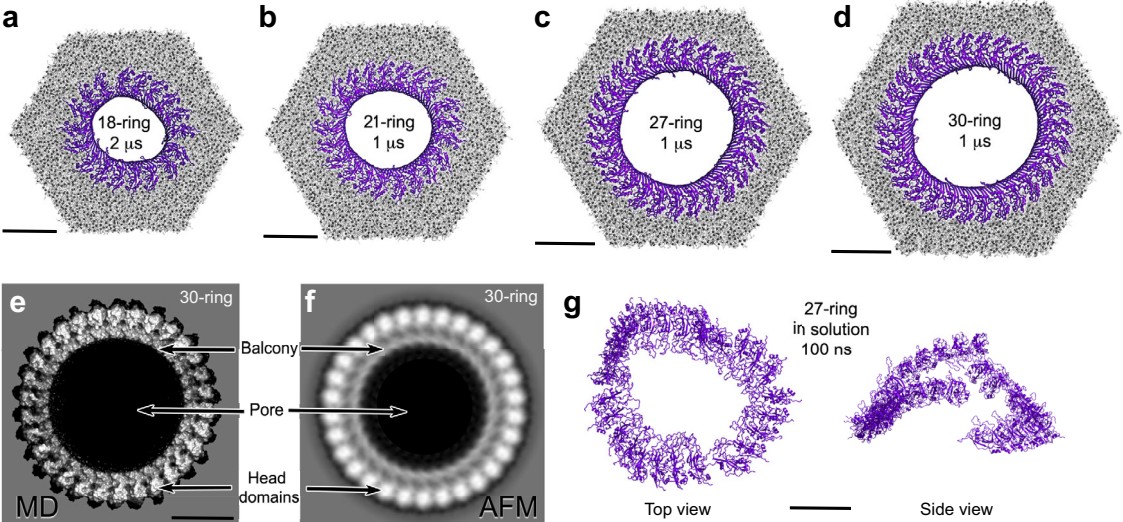

**Fig. 5 mGSDMA3^Nterm oligomers reshape into more stable assemblies. a–d** All-atom MD simulations of membrane-inserted ring-shaped oligomers encompassing 18, 21, 27 and 30 mGSDMA3^Nterm and reshaping after indicated times. Views are from cytosolic surface. *E. coli* polar lipid extract membranes are shown in gray, mGSDMA3^Nterm in purple. **e** Topograph of membrane-inserted ring-shaped oligomer of 30 mGSDMA3^Nterm averaged from 4000 oligomers from one MD simulation after exclusion of the first 100 ns. **f** Correlation symmetrized average of AFM topographs of 18 membrane-inserted ring-shaped mGSDMA3^Nterm oligomers showing 30-fold symmetry (Fig. 4d). **g** Ring-shaped oligomer without β-barrel disassembles in solution within 100 ns. Scale bars, 10 nm.

The high-resolution AFM topographs of ring-shaped mGSDMA3^Nterm oligomers showed pre-pores plugged by lipid membrane, by mGSDMA3^Nterm, or by higher protruding structures likely resulting from aggregated lipids and/or proteins (Fig. 4e). However, other oligomers showed open transmembrane pores partially or completely devoid of lipids and proteins. Neither the shape nor the size of the oligomers correlated with the formation of open pores, as small and large arc-, slit-, and ring-shaped oligomers could co-exist in both pre-pore and pore states (Figs. 1f, g and 4e). Occasionally, we observed individual mGSDMA3^Nterm oligomers transiting from the plugged pre-pore state to the open pore state, without changing the height at which they protruded above the membrane (Supplementary Fig. 6). The observations thus raised the idea that the various pre-pore conformations represent different, kinetically trapped stages towards opening the lytic transmembrane pore.

**mGSDMA3^Nterm oligomers reshape into more stable assemblies.** Next, we studied the mechanisms allowing mGSDMA3^Nterm to assemble membrane-inserted oligomers of a wide variety of stoichiometries and shapes by MD simulations. In silico models of arc-, slit-, and ring-shaped mGSDMA3^Nterm oligomers were based on the cryo-TEM structure of the ring-shaped mGSDMA3^Nterm oligomer with β-barrel[21] (Supplementary Fig. 7). The wide variety of slit- or ring-shaped mGSDMA3^Nterm oligomers remained stably inserted in the membrane during simulation times ranging from 1 to 4 µs for all-atom and up to 120 µs for coarse-grained simulations (Fig. 5a–d, Supplementary Figs. 8, 9 and Supplementary Table 2). Our MD simulations showed ring-shaped oligomers composed of 8, 12, 14, 16, 18 and 21 mGSDMA3^Nterm to reshape into elliptical shapes or slits. On the contrary, larger oligomers composed of 23, 27 and 30 mGSDMA3^Nterm retained their ring-like shape. Particularly, the atomistic ring-shaped oligomer having 27-fold stoichiometry agreed within an average root mean square deviation (RMSD) of all $C_\alpha$ atoms < 0.4 nm after simulation times of 1 µs with the cryo-TEM structure of the ring-shaped mGSDMA3^Nterm oligomer with β-barrel[21]. The RMSD, which did not increase with

simulation time, suggested the oligomer to reside in an equilibrated state. Also, visual comparison revealed that the averaged MD topograph of ring-shaped mGSDMA3^Nterm oligomers having 30-fold stoichiometry agreed excellently with the corresponding averaged AFM topograph at a resolution of ≈2 nm (Fig. 5e, f). Similar agreement between simulation and experiment was also observed for other stoichiometries of ring-shaped mGSDMA3^Nterm oligomers (Supplementary Fig. 10).

As control, we atomistically simulated ring-shaped mGSDMA3^Nterm oligomers without transmembrane β-barrel[21] in solution in the absence of a lipid membrane (Fig. 5g). The oligomers disassembled within 100 ns. In contrast, atomistic MD simulations of the ring-shaped mGSDMA3^Nterm oligomer without transmembrane β-barrel[21] adsorbed onto the lipid membrane remained stable for the time course of the simulations, thus indicating the necessity of the membrane for stabilization (Supplementary Fig. 11). These findings are in agreement with our observation of mobile membrane-attached ring-shaped mGSDMA3^Nterm oligomers (Fig. 2a–h), which can easily be washed away (Supplementary Fig. 4), and with our TEM controls showing no ring-shaped mGSDMA3^Nterm oligomers to assemble in the absence of liposomes (Fig. 3m).

Next, we analyzed the membrane contact sites of ring-shaped oligomers. The contact sites between the membrane and the head domains appeared to be the same for membrane-attached ring-shaped mGSDMA3^Nterm oligomers without β-barrel and for membrane-inserted oligomers with β-barrel (Supplementary Fig. 12). The contact sites include the positively charged α1 helix and the β1–β2 loop, which features a hydrophobic tip flanked by basic residues R41, K42, R43, K44 and R52. The positively charged α1 helix was shown to be critical for the interaction with acidic lipids[21] and the β1–β2 loop was suggested to be the initial membrane engagement site of hGSDMD[12]. In addition, the positively charged, highly conserved R18, in the loop following the α1 helix, and the highly basic flexible C-terminus of mGSDMA3^Nterm (i.e., residues K234, I235, R236 and R237) contact the negatively charged phosphates of the lipid head-groups. However, since our experiments show that only membrane-inserted mGSDMA3^Nterm oligomers form

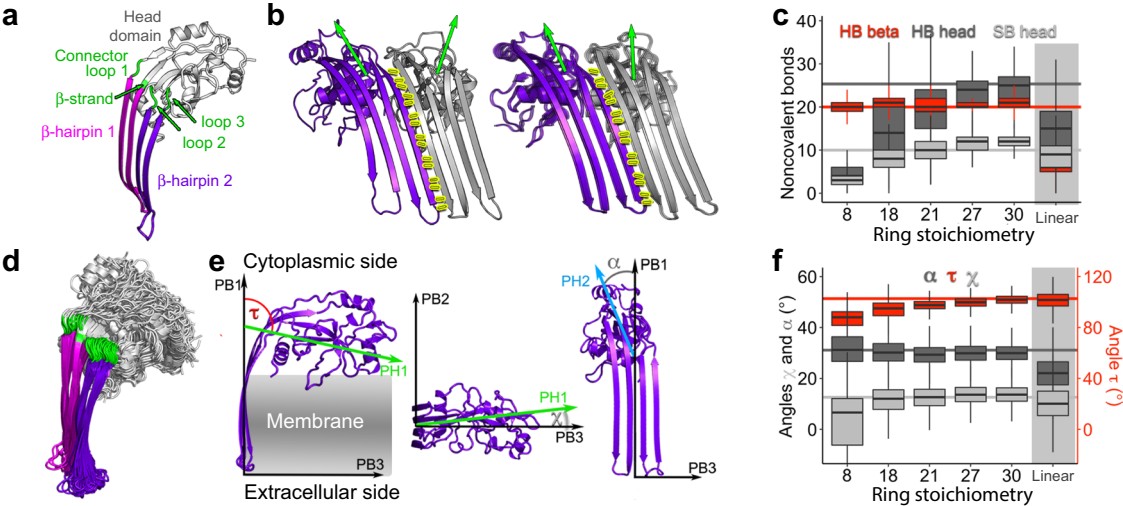

**Fig. 6 mGSDMA3$^{Nterm}$ head domain rotates to stabilize ring-shaped oligomers of different stoichiometries. a** Membrane-inserted mGSDMA3$^{Nterm}$ structure taken from Fig. 5d. Transmembrane β-hairpins are colored magenta and purple, hydrophilic head domain light gray, and 'elastic head-to-hairpin connector' green. **b** Hydrogen bond network (yellow sticks) between β-hairpins of neighboring mGSDMA3$^{Nterm}$ (colored purple and gray) in membrane-inserted ring-shaped oligomers formed by 18 (left) and 30 (right) mGSDMA3$^{Nterm}$. Green arrows highlight orientations of head domains. **c** Number of noncovalent interactions formed by each mGSDMA3$^{Nterm}$ within membrane-inserted ring-shaped oligomers having different stoichiometries. HB beta gives hydrogen bonds between β-hairpins of neighboring mGSDMA3$^{Nterm}$. HB head gives hydrogen bonds and SB head salt bridges between hydrophilic head domains. The data contains two simulations per oligomeric stoichiometry with $n > 9000$ time values taken every 100 ps for the whole simulation length after exclusion of the first 100 ns for equilibration purposes (Supplementary Table 2). **d** Flexibility of connector and head domain of membrane-inserted mGSDMA3$^{Nterm}$. Overlay of mGSDMA3$^{Nterm}$ from ring-shaped oligomers having 8, 18 and 30 stoichiometries, each simulated for 1 μs. **e** Angles describing the head domain movement of membrane-inserted mGSDMA3$^{Nterm}$. The angle τ (red) between the principal axes PH1 and PB1 describes the vertical movement of the head domain along the membrane normal. The angle χ (light gray) between the principal axes PH1 and PB3 describes the horizontal movement of the head domain along the membrane plane. The angle α (dark gray) describes the rotation of the second principal axis of the head domain (PH2) around the first principal axis of the β-sheets (PB1). **f** Rotation of membrane-inserted mGSDMA3$^{Nterm}$ head domains in dependence of the ring-shaped oligomer stoichiometry. Angles α, τ and χ as in (**e**). Data follows that in (**c**). Horizontal lines in (**c, f**) visualize the bonds and angles of the in vacuo energy minimized cryo-TEM structure of the membrane-inserted ring-shaped oligomer assembled from 27 mGSDMA3$^{Nterm[21]}$. Boxplot visualizations follow standard R settings, i.e. bottom and top of the box are 25th (Q1) and 75th (Q3) percentile, the band near the middle of the box is the median, the whiskers extend the box size (Q3–Q1) 1.5 times.

transmembrane lytic pores, we in the following focus our MD simulations on membrane-inserted oligomers.

**Head domain and β-hairpins stabilize oligomers complementarily.** We next analyzed at atomic scale the structures and interactions of mGSDMA3$^{Nterm}$ assembling ring-shaped oligomers of different stoichiometries in lipid membranes (Fig. 6). Irrespective of the stoichiometry, the transmembrane β-hairpins from adjacent mGSDMA3$^{Nterm}$ established the same network of hydrogen bonds as observed for the cryo-TEM structure of the ring-shaped oligomer composed of 27 mGSDMA3$^{Nterm}$ (Fig. 6a–c). Thus, the hydrogen bonds formed between the β-hairpins contribute fundamentally to the stabilization of mGSDMA3$^{Nterm}$ oligomers. However, with increasing stoichiometry, mGSDMA3$^{Nterm}$ increased the tilt of the hydrophilic head domain relative to the β-hairpins (Fig. 6d–f). The 'elastic head-to-hairpin connector' enabling this tilt comprises four polypeptide chains, one of which adopting a β-stranded conformation and three remaining unstructured (Fig. 6a, d, Supplementary Fig. 13). By this structural adaptation of the head domains, mGSDMA3$^{Nterm}$ maximized the number of hydrogen bonds and salt bridges to neighboring head domains while preventing structural clashes. Consequently, the number of hydrogen bonds and salt bridges formed between the head domains in larger ring-shaped oligomers of 27- or 30-stoichiometry was similar to that observed in the cryo-TEM structures of the ring-shaped oligomers with β-barrel (Fig. 6c)[21]. However, smaller ring-shaped oligomers could not establish stabilizing interactions between all

hydrophilic head domains and reshaped into elliptic oligomers (Fig. 5a, b, Supplementary Figs. 8, 9). This observation agrees with our AFM topographs (Figs. 1a–e, 4a), which show small arc- and slit-shaped oligomers instead of ring-shaped oligomers having less than 18 mGSDMA3$^{Nterm}$.

The number of intermolecular hydrogen bonds and salt bridges that increases with the stoichiometry of ring-shaped oligomers indicates that larger membrane-inserted mGSDMA3$^{Nterm}$ oligomers increase in stability (Fig. 6c). However, our experiments show that the population of ring-shaped oligomers containing ≥ 32 mGSDMA3$^{Nterm}$ decreases rapidly (Fig. 4b). To better understand this size limitation, we simulated membrane-inserted ring-shaped oligomers of 40 mGSDMA3$^{Nterm}$ at coarse-grained resolution and linear-shaped mGSDMA3$^{Nterm}$ oligomers, corresponding to a ring of infinite curvature, at all-atom resolution (Supplementary Fig. 14). During equilibration all oligomers attempted to maximize the number of stabilizing interactions between β-hairpins and head domains of adjacent mGSDMA3$^{Nterm}$. However, mGSDMA3$^{Nterm}$ could not rearrange the head domains sufficiently to avoid steric clashes and simultaneously maintain the stabilizing interactions between β-hairpins (Fig. 6c, f). Consequently, the hydrogen bonding between β-hairpins of some neighboring mGSDMA3$^{Nterm}$ disrupted and caused the large oligomers to reshape into smaller arc-shaped oligomers (Supplementary Fig. 14).

**Monitoring mGSDMA3$^{Nterm}$ oligomers forming lytic pores.** Coarse-grained simulations of arc-, slit- and ring-shaped

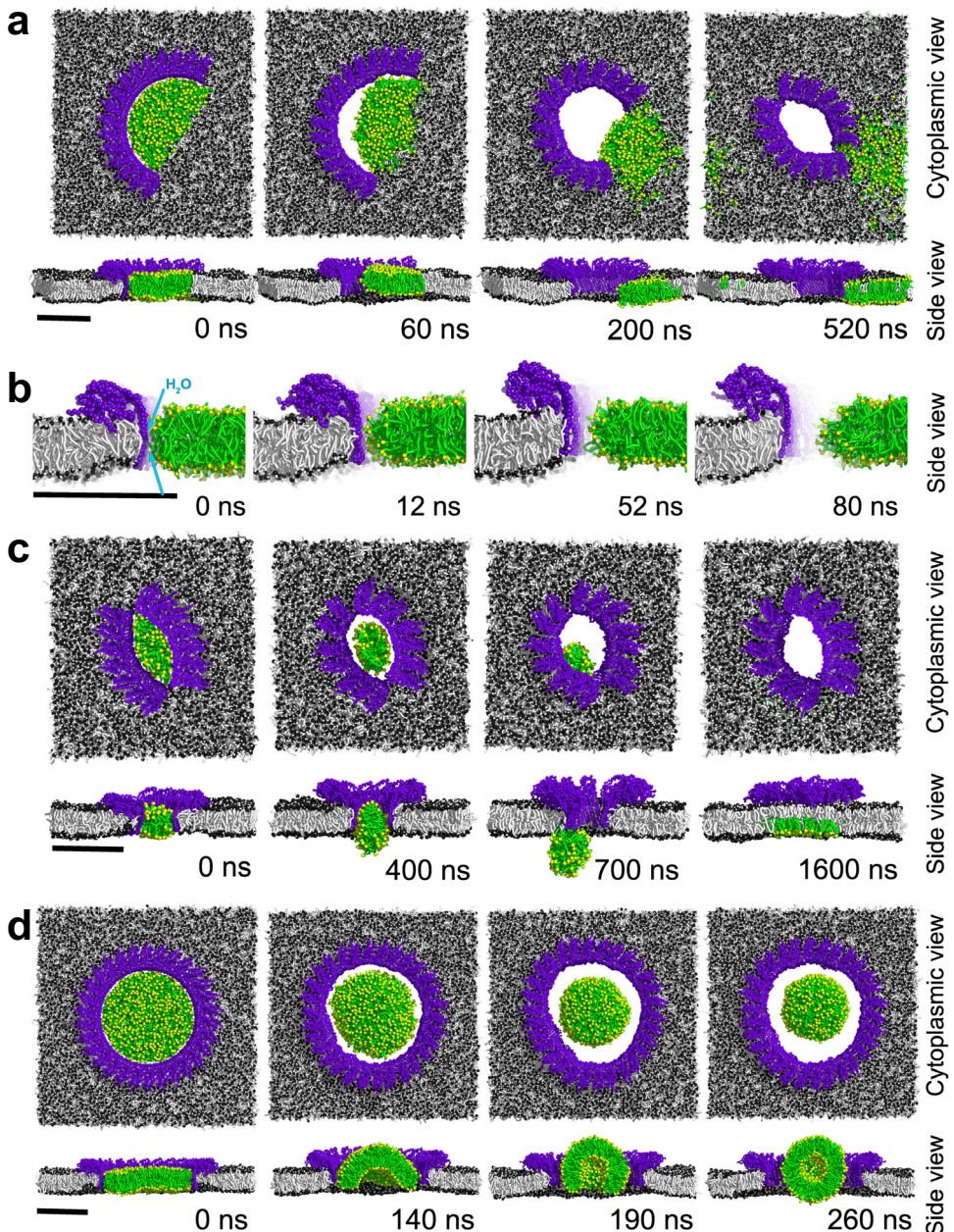

**Fig. 7 Pore formation of membrane-inserted mGSDMA3[Nterm] oligomers. a** Pore formation of an arc-shaped oligomer assembled from 16 mGSDMA3[Nterm] ($n = 8/8$ simulations). **b** Lipid withdrawal from mGSDMA3[Nterm] forming the arc-shaped oligomer shown in (**a**) at higher time resolution and structural detail. The hydrophilic (polar) surface of the transmembrane β-sheet formed by adjacent mGSDMA3[Nterm] causes water to penetrate into the membrane. The intrusion of water along the β-sheet is followed by the reassembly of lipid headgroups (yellow), whose withdrawal from the hydrophilic surface opens the transmembrane pore. **c** Pore formation of a slit-shaped oligomer assembled from 14 mGSDMA3[Nterm]. The lipids of the pre-pore fuse into the surrounding membrane (observed in 1 of 5, $n = 1/5$ simulations). **d** Pore formation of a ring-shaped oligomer assembled from 30 mGSDMA3[Nterm]. Lipids left ring-shaped oligomers assembled from 27 ($n = 7/7$ simulations) or 30 ($n = 6/6$ simulations) mGSDMA3[Nterm] as vesicles. mGSDMA3[Nterm] are colored in purple, lipid fatty acid chains light gray, and lipid headgroups dark gray. Lipids withdrawing from the transmembrane β-sheet thereby forming the transmembrane pore are colored green with yellow headgroups. The top row in each subfigure shows the cytoplasmic view, the bottom row the side view. Simulation times are indicated for each snapshot. Scale bars, 10 nm. The entire simulations are shown in Supplementary Movie 2–10. Additional coarse-grained MD simulations of arc-, slit- and ring-shaped oligomers forming transmembrane pores are shown in Supplementary Fig. 15.

mGSDMA3[Nterm] oligomers, whose pre-pores were plugged with lipid membrane allowed the pore formation process to be monitored (Fig. 7, Supplementary Fig. 15 and Movie 2–10). Because the transmembrane β-hairpins lining the pre-pore are hydrophobic on the outside and hydrophilic on the inside (Supplementary Fig. 16)[21], the simulations showed the hydrophilic inner surface of the β-stranded pre-pore to draw water into the protein-lipid interface (Fig. 7a, b). This process forced the lipids to

withdraw until they bulged and formed vesicles or nanodiscs that exited into the solution and opened the lytic transmembrane pore (Fig. 7, Supplementary Fig. 15 and Movie 2–10). Occasionally, the lipids inside oligomeric pre-pores bulged and fused with the surrounding membrane. Preferentially, the lipids left the pre-pore of small sized oligomers via membrane fusion (35,7%; $n = 5/14$) or nanodiscs (50%; $n = 7/14$) and less often as vesicles (14.3%; $n = 2/14$), while they left large ring-shaped oligomers always as

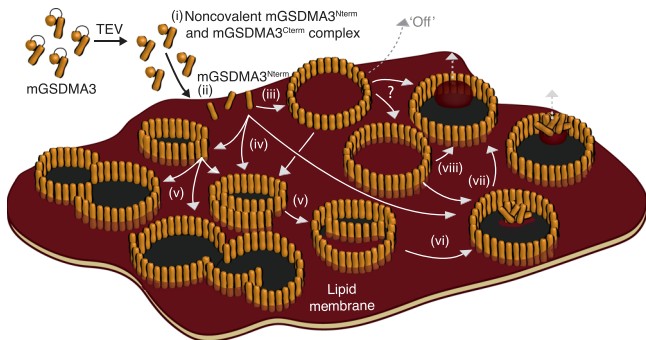

**Fig. 8 Pathways of mGSDMA3^Nterm membrane pore formation.**
(i) mGSDMA3 cleaved by the TEV protease forms the noncovalent mGSDMA3^Nterm and mGSDMA3^Cterm complex[8]. (ii) Once attached to the membrane, mGSDMA3^Nterm assembles oligomers (iii) on the lipid membrane surface and (iv) inserted into the lipid membrane. (iii) Mobile membrane-attached ring-shaped mGSDMA3^Nterm oligomers are weakly adsorbed onto the membrane and can be rinsed off by exchanging the buffer solution ('Off'). (iv) Relatively immobile membrane-inserted slit- and arc-shaped oligomers grow (v-vi) larger arcs, arc clusters, and rings by the addition of mGSDMA3^Nterm to their free ends. Oligomers not yet having opened their transmembrane pore can be considered to reside in the pre-pore state whereas oligomers having opened their pore reside in the pore state. (vi-viii) In all membrane-inserted oligomeric shapes, the hydrophilic inner surface of the transmembrane β-sheets or β-barrels attracts water and repels the lipid membrane, which opens the transmembrane pore. During this process, lipids leave the oligomer either by fusing with the surrounding membrane or by forming micelles, nanodiscs or vesicles. During the entire process, membrane-attached and membrane-inserted mGSDMA3^Nterm oligomers residing in the pre-pore or pore state protrude by the same height above the lipid membrane, indicating the absence of a vertical collapse.

vesicles (100%; $n = 13/13$). In all simulations, mGSDMA3^Nterm oligomers transitioned from the pre-pore to the lytic pore state without significant changes in height, which is in full agreement with the analysis of the AFM data (Figs. 1, 2, 4 and Supplementary Fig. 6).

## Discussion

**Membrane-attached and membrane-inserted mGSDMA3^Nterm oligomers.** Upon incubating lipid membranes with mGSDMA3^Nterm we observe the assembly of two main classes of oligomers. One class represents mobile ring-shaped oligomers, which do not form transmembrane pores (Fig. 8). The mobile oligomers can be washed away by rinsing the membrane with buffer solution. The ease of removal shows that the mobile ring-shaped oligomers are weakly adsorbed on the membrane and not membrane-inserted. These mobile oligomers resemble the recently reported cryo-TEM structures of solubilized ring-shaped mGSDMA3^Nterm and hGSDMD^Nterm oligomers, which have no β-barrel and were suggested to represent possible pre-pore conformations[12,21]. Our experiments and simulations show that such oligomers need the lipid membrane to assemble and stabilize. However, the exact mechanism by which soluble ring-shaped oligomers insert into membranes remains unknown.

The other class represents relatively immobile arc-, slit-, and ring-shaped oligomers, which are membrane-inserted and can exist in either the plugged pre-pore or open pore state (Fig. 8). In contrast to mobile membrane-attached ring-shaped oligomers, the membrane-inserted oligomers cannot be washed away and remain stably inserted and assembled in the membrane for several days. The ability of arc- or slit-shaped oligomers, in addition to ring-shaped oligomers, to perforate the membrane,

has also been described for hGSDMD[9,22], perforin[28,29], Bax[30], and the cholesterol-dependent cytolysins (CDCs) suilysin[31], listeriolysin O[32], pneumolysin[33,34] and streptolysin O[35–38]. Both the wide size distribution and several days long coexistence of arc-, slit-, and ring-shaped membrane-inserted mGSDMA3^Nterm oligomers indicate that the self-assembly of mGSDMA3^Nterm processes along variable pathways with different termination points.

**mGSDMA3^Nterm oligomers transit into pore state without vertical collapse.** Our experiments and simulations capture membrane-inserted mGSDMA3^Nterm oligomers transiting from the pre-pore to the pore state. In vivo, the osmotic pressure of the cell may accelerate the expelling of lipids and/or proteins plugging the pre-pores. The observation that mobile membrane-attached ring-shaped oligomers, as well as the membrane-inserted arc-, slit-, and ring-shaped mGSDMA3^Nterm oligomers residing in either the pre-pore or the pore state, all protrude by same height from the membrane shows that neither the insertion of the oligomer into the membrane nor the formation of the lytic pore requires a vertical collapse (Fig. 8). The absence of a vertical collapse agrees with the cryo-TEM structures, which show that the head domains of ring-shaped oligomers of mGSDMA3^Nterm and hGSDMD^Nterm without and with β-barrel have comparable heights[12,21]. The absence of vertical collapse of membrane-inserted mGSDMA3^Nterm oligomers upon transiting from the pre-pore to the pore state has also been reported for hGSDMD^Nterm,[22] while pore formation associated with the vertical collapse of a pre-pore has been described for the CDCs perfringolysin O[39,40], suilysin[31] and pneumolysin[41,42]. Remarkably, the sequences and 3D-structures of gasdermins differ significantly from any other known pore-forming protein[6,21]. Given the conserved lytic pore-forming activity of gasdermins, the formation of membrane-attached and membrane-inserted oligomers, the flexible assembly of arc-, slit-, and ring-shaped oligomers, and the pre-pore to pore transition in the absence of vertical collapse are thus likely to be general features of gasdermin family members.

**Membrane-inserted and membrane-attached oligomerization of mGSDMA3^Nterm.** Time-lapse AFM shows membrane-inserted mGSDMA3^Nterm oligomers assembling small slit- and arc-shaped oligomers that gradually grow into larger oligomers and oligomeric clusters. In agreement with previous findings on hGSDMD[22], oligomerization and pore formation are initiated in the lipid membrane and feature a continuous growing process without a vertical collapse of pore-forming proteins[43]. This scenario describes a mode of membrane-inserted oligomerization (Fig. 8). An alternative mode is the membrane-attached oligomerization, in which oligomers assemble on the membrane surface and only afterwards insert into the membrane. Our time-lapse AFM experiments also show mobile membrane-attached rings, as they would occur in the membrane-attached oligomerization mode. These mobile rings have the same height above the membrane as the membrane-inserted pre-pores and pores. The mobile membrane-attached mGSDMA3^Nterm ring-shaped oligomers resemble the cryo-TEM structures of ring-shaped oligomers without β-barrel of mGSDMA3^Nterm and hGSDMD^Nterm, which were speculated to represent pre-pore conformations[12,21]. Altogether, time-lapse AFM experiments provide evidence that both membrane-attached and membrane-inserted oligomerization pathways co-exist for gasdermins.

**Structural mechanisms stabilizing oligomers and opening pores.** Our high-resolution AFM topographs show differently

shaped and sized membrane-inserted mGSDMA3[Nterm] oligomers, which are stable over the time course of days. Atomistic and coarse-grained MD simulations of these oligomers, which were based on the mGSDMA3[Nterm] structure[21], were used to explore the intermolecular interactions stabilizing the observed arc-, slit- and ring-shaped oligomers. The network of hydrogen bonds stabilizing the transmembrane β-sheets and β-barrels largely remains unchanged in all oligomeric assemblies studied, thus demonstrating that the transmembrane β-hairpins can facilitate a wide variety of oligomeric sizes and shapes. However, in order to further stabilize the various oligomeric sizes and shapes, the mGSDMA3[Nterm] head domains tilt parallel and perpendicular to the membrane plane, which requires considerable structural plasticity. The plasticity of the head domain is facilitated by a structurally flexible region connecting the head domain with the transmembrane β-hairpins. Naturally, the movements of the head domain are limited in adapting to too small or too large ring-shaped oligomers. These limitations explain why we experimentally observe mGSDMA3[Nterm] oligomers only within a certain size range.

To open the lytic transmembrane pore the transmembrane β-sheets or β-barrels of mGSDMA3[Nterm] oligomers attract water along the hydrophilic β-stranded inner surface of the pre-pore and force the lipids plugging the pre-pore to withdraw, similarly as observed in recent MD simulations of pneumolysin[44]. Withdrawing lipids then bulge and either leave the pore in form of micelles, nanodiscs, or vesicles or fuse with the membrane surrounding the oligomer (Fig. 8). These two pathways reflect the two lipid removal mechanisms termed vertical expulsion (by micellation or vesiculation) and lateral retreat[42,44]. Interestingly, both pathways have been described for pneumolysin forming β-stranded pores > 30 nm in diameter[42,44], and for cytolysin A forming α-helical pores < 10 nm in diameter[45]. These observations together with our results obtained for mGSDMA3[Nterm] suggest a generality: Lipids can be removed along different pathways from the proteinaceous pore, regardless of the type of the protein (large β-stranded pneumolysin, medium-sized β-stranded gasdermin, or small α-helical cytolysin), as long as the inner wall of the pore is hydrophilic. This supposition is corroborated by the fact that AFM imaging of mGSDMA3[Nterm], pneumolysin[42] and listeriolysin O[32] recorded similar transitions from the plugged pre-pore to the lytic, open pore state with occasionally kinetically trapped intermediate states. One may further speculate that the structural flexibility of gasdermins to assemble pore-forming oligomers of various shapes and sizes represents an advantageous mechanism to insert pores in any given biological context and under external constraints such as imposed in densely packed cellular membranes, which dynamically adapt their biochemical and mechanical properties to the cell state[46–48]. At the same time, the limited structural plasticity of mGSDMA3[Nterm] encodes a maximal diameter of the oligomeric pore, which provides a size limit for the diffusive export of biomacromolecules during pyroptosis.

In summary, we here observe the assembly of mGSDMA3[Nterm] oligomers that are either mobile and weakly attached on the membrane surface or immobile and stably inserted in the lipid membrane (Fig. 8). Although we could not monitor membrane-attached oligomers inserting into the membrane and forming transmembrane pores, which may reflect technical limitations, their mobility on the membrane surface may help them find a suitable position to insert into the membrane. However, once inserted, the oligomers assemble and grow a variety of oligomeric shapes and sizes, each able to transit from the plugged pre-pore state to the open pore state by the expulsion of lipids (and proteins) from the pre-pore. The ability of mGSDMA3[Nterm] to assemble stable oligomeric pores of diverse shapes and sizes is

determined by two distinct structural elements, the formation of a large transmembrane β-sheet or β-barrel and the stabilization of the oligomers by the structural plasticity of the head domain. The multiple pathways along which gasdermin can assemble and open lytic pores occur in the absence of a vertical collapse. It may be speculated that this kinetic network of pore formation ensures a robust cellular response in pyroptosis.

## Methods

**Cloning, expression, and purification of mGSDMA3 and TEV.** The gene coding for mouse GSDMA3 was cloned into a pET28b expression vector, together with an N-terminal His$_6$-SUMO-tag. Full-length mGSDMA3 with the insertion of a tobacco etch virus protease (TEV) cleavage site immediately after R237 was overexpressed in BL21(DE3) E. coli strains. Cells were harvested and disrupted by high-pressure microfluidization. SUMO-mGSDMA3 was isolated by Ni$^{2+}$-NTA affinity purification and the SUMO-tag was cleaved by ULP1 protease. The protein was further purified by ion-exchange and size-exclusion chromatography. TEV was cloned, expressed in inclusion bodies in E. coli and refolded as described[49]. Briefly, inclusion bodies were solubilized in 8 M urea, and the TEV was refolded by rapid dilution. During this refolding the enzymes were auto-activated. A gel filtration column was run to purify the final product and the concentration of active TEV determined by active site titration. mGSDMA3 and TEV samples were aliquoted, flash-frozen and stored at –80 °C. Aliquots were then used only once, just after allowing the sample to thaw on ice.

**Liposome preparation.** Unilamellar liposomes (lipid vesicles) were prepared by hydration of lipid films followed by extrusion through polycarbonate filter membranes. E. coli polar lipid extract (EPL, composed of phosphatidylethanolamine, phosphatidylglycerol and cardiolipin, 67:23.2:9.8 wt/wt%, Avanti Polar Lipids) and extruding equipment used for liposome preparation were purchased from Avanti Polar Lipids. To prepare liposomes, the EPL chloroform stock solution was taken and the chloroform was evaporated under a gentle stream of nitrogen followed by overnight dehydration under a vacuum. Lipids were then rehydrated in buffer solution (150 mM NaCl, 20 mM Hepes, pH 7.4) and extruded 11 times through a 100 nm pore-size membrane. Freshly prepared liposomes (10 mg ml$^{-1}$) were immediately aliquoted, flash-frozen and stored at −80 °C. Liposomes were prepared at room temperature (RT, ≈22 °C) with the only exception of the rehydration, which was done at 42 °C for 1 h with continuous agitation at 2 x g (Thermomixer Comfort, Eppendorf). Buffer solutions were freshly made using nanopure water (18.2 MOhm cm$^{-1}$) and pro analysis (>98.5%) purity-grade reagents from Sigma-Aldrich and Merck.

**Preparation of liposomes and defect-free supported lipid membranes (SLMs).** Supported lipid membranes (SLMs) were prepared by fusion of unilamellar liposomes on mica. An appropriate volume (≈ 11 µl) of liposome solution of EPL was sonicated (25 kHz, 500 W, Transsonic TI-H-5, ELMA) for 5 min at RT after the addition of 6 µl of Ca$^{2+}$-containing buffer (150 mM NaCl, 40 mM CaCl$_2$, 20 mM Hepes, pH 7.4) and then immediately adsorbed onto a freshly cleaved mica disk of 6 mm diameter at RT[50]. After an adsorption time of ≈ 45 min, during which the liposomes fused into a SLM[32], the sample was gently rinsed with imaging buffer solution (50 mM NaCl, 0.1 mM EDTA, 0.1 mM TCEP, 20 mM Hepes, pH 7.4). SLMs were prepared freshly and imaged by AFM before being incubated with the proteins to ensure that the SLMs were defect-free and completely covered the AFM support (Supplementary Fig. 1). Only if the SLM covering the mica support showed no defects such as holes, cracks or stacked membranes, mGSDMA3 and TEV were applied. This quality criterion ensured that we observe the interaction of the proteins with an intact lipid membrane. To avoid sample preparation artefacts, the SLM was never exposed to air and the AFM head was never moved until the end of the experiment.

**Incubation of SLMs with mGSDMA3.** After AFM imaging and verifying that the SLMs prepared were defect-free, they were incubated with 1.5 µM mGSDMA3 and 0.4 µM TEV in imaging buffer solution (50 mM NaCl, 0.1 mM EDTA, 0.1 mM TCEP, 20 mM Hepes, pH 7.4) at 37 °C. Before incubation onto the SLM, the solution of 1.5 µM mGSDMA3 was pre-cleaved by 0.4 µM TEV overnight at 37 °C. After an incubation time of 3 h the SLMs were rinsed and imaged in imaging buffer solution using AFM at room temperature. For time-lapse AFM, SLMs were incubated with the pre-cleaved mGSDMA3 described above (1.5 µM mGSDMA3 pre-cleaved overnight with 0.4 µM TEV at 37 °C in imaging buffer solution) in the AFM fluid cell at 37 °C. AFM images of a selected area of the SLM were then acquired by FD-based AFM every ≈ 10–30 min for up to 5 h at 37 °C in pre-cleaved mGSDMA3 solution. Every experimental condition was reproduced at least three times using freshly prepared SLMs, new protein samples, AFM supports, and cantilevers.

**Atomic force microscopy (AFM).** Contact mode AFM was performed with a Nanoscope IIe (Veeco) equipped with a 100 µm piezoelectric scanner, a fluid cell

and oxide-sharpened Si₃N₄ cantilevers (ORC8-PS-W, Olympus), having a nominal spring constant of 0.05 N m⁻¹. AFM topographs were recorded in imaging buffer solution (50 mM NaCl, 0.1 mM EDTA, 0.1 mM TCEP, 20 mM Hepes, pH 7.4). The scanning frequency was between 2 and 8 lines s⁻¹, and the force applied to the AFM cantilever was kept < 50 pN to prevent structural perturbation of the sample[25]. Imaging forces of the AFM were manually adjusted to compensate for thermal drift. Proportional and integral gains of the AFM were adjusted manually to minimize the error (deflection) signal and to maximize the height signal. AFM topographs (512 × 512 pixels) were recorded in trace and retrace scanning directions. Topographs were analyzed using the AFM analysis software (NanoScope Analysis 1.8).

FD-based AFM[26,27] was performed with a Nanoscope Multimode 8 (Bruker, USA) operated in PeakForce Tapping mode. The AFM was equipped with a 120 µm piezoelectric scanner and fluid cell. The two different AFM cantilevers used either had a nominal spring constant of 0.1 N m⁻¹, resonance frequency of ≈ 110 kHz in liquid and a sharpened silicon tip with a nominal radius of 8–10 nm (BioLever mini BL-AC40, Olympus Corporation) or a spring constant of 0.4 N m⁻¹, a resonance frequency of ≈ 165 kHz in liquid and a sharpened silicon tip with a nominal radius of ≈ 1 nm (PEAKFORCE-HiRs-F-A, Bruker Nano Inc., USA). BL-AC40 AFM cantilevers were calibrated using thermal tuning method[51] and by ramping the cantilever on the mica support. To prevent the damage of the ultra-sharp tip having a nominal radius of ≈ 1 nm, we used the calibration provided for PEAKFORCE-HiRs-F-A cantilevers. AFM topographs were recorded in imaging buffer solution (50 mM NaCl, 0.1 mM EDTA, 0.1 mM TCEP, 20 mM Hepes, pH 7.4) as described[52]. Briefly, the maximum force applied to image the samples was limited to ≈ 100 pN and the oscillation frequency and oscillation amplitude of the cantilever were set to 2 kHz and 25 nm, respectively. Topographs were analyzed using the AFM analysis software (NanoScope Analysis 1.8).

Both, contact mode and FD-based, AFMs were placed inside a home-built temperature controlled acoustic isolation box. AFM topographs were recorded at RT with the exception of the time-lapse AFM experiments, which were recorded at ≈ 37 °C.

**AFM image analysis.** Averages of trace and retrace AFM topographs were calculated and drift corrected (unwarped). Quantitative unwarping was possible with images that exhibited drift only along the y-(slow scan) axis, permitting the pixel size calibration of the x-(fast scan) axis to be applied. Well-preserved ring-shaped oligomers were then manually selected for stoichiometry determination, correlation averaging and rotational symmetrization. Rotational power spectra of 193 rings were calculated of which 174 exhibited a distinct major peak that allowed rings to be classified into groups with identical stoichiometry. The diameters of rings from quantitatively unwarped topographs were found to have a distribution with discrete peaks that related to the ring stoichiometry. This allowed the pixel size to be calibrated with the atomic model of the ring-shaped oligomer. All ring-shaped oligomers within these classes were then scaled with respect to rings from the quantitatively unwarped AFM topographs. Scaled ring-shaped oligomers were averaged after rotational and translational alignment to a reference for several cycles, selecting the calculated average as reference for the next cycle, eliminating rings that had a correlation <0.9 with the reference.

**Transmission electron microscopy (TEM).** For negative-stain TEM, 5 µl of sample was pipetted onto a glow-discharged copper grid coated with parlodion and carbon and left to adsorb for 1 min at RT. The grid was then washed with 4 droplets of nanopure water, and subsequently stained with 2 % uranyl acetate for 10 s, blotting between each step. Grids were scanned using a Tecnai G2 Spirit TEM microscope with a LaB6 filament operated at 120 kV (FEI Company, Eindhoven, The Netherlands). Images were recorded by a side-mounted EMSIS MORADA camera.

**Molecular dynamics (MD) simulations**
*Preparation of monomeric mGSDMA3^Nterm.* Monomeric structures of mGSDMA3^Nterm were based on the cryo-TEM structure of the ring-shaped mGSDMA3^Nterm oligomer 6CB8[21]. Missing residues 66-PGSS-69 and 234-KIRR-237 were added by Modeller[53]. Both mGSDMA3^Nterm termini were chosen to be charged and titratable residues carried charges corresponding to their standard protonation state at physiological pH. This structure was energy minimized in vacuo by 10,000 steps of steep-descent algorithm and served as a building block for preparation of mGSDMA3^Nterm oligomers.

*Construction of mGSDMA3^Nterm oligomers.* Membrane-inserted ring-shaped oligomers were prepared by rotating the above prepared mGSDMA3^Nterm monomer N times by 360/N degrees. By varying the distance of mGSDMA3^Nterm from the ring center, the optimal ring radius was determined based on the minimum of the total potential energy of the oligomer in vacuo after energy minimization. Arc-shaped oligomers were generated by cutting a certain number of mGSDMA3^Nterm from ring-shaped oligomers. Slit-shaped oligomers were generated by placing two arc-shaped oligomers next to each other. Each oligomer was energy minimized in vacuo by 10,000 steps of steep-descent algorithm. In case of the fully solvated ring-shaped oligomer containing 27 mGSDMA3^Nterm, the β-hairpins in the cryo-TEM structure 6CB8[21] were remodeled as loops by Modeller[53].

*Simulation conditions.* All sequential multiscaling MD simulations[54] were performed at 37 °C and a pressure of 1 bar in GROMACS version 2018.x[55]. As a membrane model a free standing symmetric membrane bilayer made of *E. coli* polar lipids extract has been used[56]. This model contained 14 different lipid species including cardiolipin, phosphatidylethanolamine, and phosphatidylglycerol headgroups and unsaturated, saturated and cyclopropanylated tails in proportions corresponding to a typical in vitro sample.

*Coarse-grained simulations.* Energy minimized mGSDMA3^Nterm oligomers were converted to coarse-grained resolution of Martini 3 force field[57] using *martinize3* (kindly obtained from S.-J. Marrink) and energy minimized by performing 10,000 steps of steepest descent energy minimization. Next, one cardiolipin was added to each mGSDMA3^Nterm at a binding position suggested by cryo-TEM[21] and the lipids were energy minimized while keeping mGSDMA3^Nterm frozen. Afterwards, the *E. coli* polar lipid extract membrane, nonpolarizable water and 100 mM NaCl were added using *insane*[58]. The resulting simulation system was energy minimized twice. First, for 1000 steps mGSDMA3^Nterm structures were kept frozen and then for 10,000 steps all particles were allowed to move freely. Afterwards, a set of equilibration simulations with positional restrains on the proteins and increasing time steps (i.e. 2 fs, 5 fs, 10 fs) were performed for 100 ps, 50 ps, and 1000 ps, respectively. One additional MD simulations using the 20 fs time step and lasting for 1 ns was performed with positional restrains on the backbone in order to allow for adaptation of protein sidechains. Production run simulations were performed using 20 fs timesteps for at least 1 µs.

In all coarse-grained simulations (summarized in Supplementary Table 2) the temperature was controlled by v-rescale thermostat to be at 37 °C, using $\tau_t = 1$ ps. Pressure was kept at 1 bar by the Berendsen barostat and $\tau_p = 12$ ps in semiisotropic manner, using the compressibility of $3 \times 10^{-4}$ bar⁻¹. Electrostatics was described by reaction-field within 1.1 nm with $\varepsilon_r$ of 15. Van der Waals interactions were cut-off at 1.1 nm using the potential-shift-Verlet van der Waals modifier[59]. The secondary structure within the β-sheets and within the mGSDMA3^Nterm hydrophilic head domain was separately stabilized by RubberBands[60] using the decay power $p = 3$ and decay factor $a = 1$, force constant of 50 kJ mol⁻¹ nm⁻¹ and distance 1.5 nm. Thus, mGSDMA3^Nterm hydrophilic head domain and transmembrane β-sheets can move independently while retaining their secondary structure.

*All-atom simulations.* Coarse-grained structures were converted back to all-atom resolution using CHARMM36m force field[61,62] by *backward*[63]. The CHARMM36 force field was chosen because it includes parameters for diverse lipid species and because it was thoroughly tested to describe protein-lipid interactions in good agreement with experimental data[64,65]. After a fit of the energy minimized cryo-TEM structure, the atomistic system was energy minimized twice. First, for up to 20,000 steps while keeping the protein frozen, second, for up to 5000 steps letting all atoms to move freely. Next, velocities were generated and a short simulations of 5000 steps using a 0.2 fs time step with position restraints on the protein was performed using both the Berendsen[66] thermostat (37 °C, $\tau_t = 0.5$ ps) and barostat (semiisotropic pressure coupling, 1 bar, $\tau_p = 5$ ps, compressibility $4.5 \times 10^{-5}$ bar⁻¹). Then a 20 ns long simulation using the 2 fs time step with position restraints on all non-hydrogen atoms was performed to relax the surroundings, followed by a 20 ns simulation with position restraints on protein backbone heavy atoms, allowing the sidechains and the surrounding to co-adjust. Finally, production run simulations were started from the last frame. In all-atom simulations (summarized in Supplementary Table 2) the TIP4P water model[67] was used as recommended for CHARMM force field[68] in GROMACS[68]. The temperature of 37 °C was controlled by the Nosé-Hoover thermostat[69] using a time coupling constant of 0.5 ps, the pressure was controlled in semiisotropic manner by the Parrinello-Rahman[70,71] barostat to be at 1 bar, with a time constant of 1 ps and compressibility of $4.5 \times 10^{-5}$ bar⁻¹. The bonds to hydrogen atoms were constrained using LINCS[72], the neighbor list was updated every 10 steps using the Verlet cut-off scheme[73] and buffer tolerance of 0.005 kJ mol⁻¹ ps⁻¹ per atom. The electrostatics behind 1.2 nm was described by particle-mesh-Ewald using the Potential-shift-Verlet modifier and the van der Waals interactions were shifted to zero between 0.8 and 1.2 nm using the Potential-switch method.

A list of all performed simulations is shown in Supplementary Table 2. The analysis succeeded by standard GROMACS tools and home written scripts in python 3[74] and R 4[75]. The images of the molecules were generated using PyMOL 2.4[76] and the plots by R 4[75].

**Statistics and reproducibility.** Statistical analysis and computing for the experimental and theoretical results were applied as described. Every AFM experiment described has been reproduced at least three independent times using different mGSDMA3 samples, AFM cantilevers, AFM supports, SLMs and different AFMs. Some of the AFM experiments (e.g., Figs. 1, 2, 4, Supplementary Figs. 1–5) have been reproduced more than 10 times over the time course of several months. The TEM experiments have been reproduced at least three independent times, each time using freshly prepared samples. Each of the experimental replicates provided similar results. Each MD simulation of a total of 26 different systems has been typically reproduced at least two independent times. A summary of all simulations

is given in Supplementary Table 2. Every theoretical replicate provided similar results.

**Reporting summary**. Further information on research design is available in the Nature Research Reporting Summary linked to this article.

## Data availability
The experimental and simulated datasets produced in this study are available from the corresponding authors upon reasonable request.

## Code availability
R script code evaluating the tilt and rotation angles of the gasdermin head domain relative to the transmembrane β-hairpins is available upon reasonable request.

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

## Acknowledgements

K.P. acknowledges financial support of the German Research Foundation (DFG) for Forschungsstipendium PL 853/1-1 and computing power grant from the Swiss National Supercomputing Centre (CSCS) on Piz Daint under project ID s945. This work was supported by the Swiss National Science Foundation (SNSF) of the National Competence Centre in Research (NCCR) Molecular Systems Engineering. We kindly acknowledge the assistance of K. Kasuba towards determining the head rotational movement α. Moreover, we kindly acknowledge S.-J. Marrink and P. Telles de Souza for sharing Martini 3 parameters with us well prior their publication.

## Author contributions

Experiments and simulations were designed by S.A.M., K.P. and D.J.M. mGSDMA3 samples were prepared by J.P. and S.H. AFM and TEM imaging was conducted by S.A.M. AFM experiments were analyzed by S.A.M, M.L. and A.E. MD simulations were conducted and analyzed by K.P. All authors discussed and wrote the manuscript.

## Competing interests

The authors declare no competing interests.
