## [Peer Review File · Nature Communications]

Gasdermin-A3 pore formation propagates along variable pathwaysReviewers' Comments:

Reviewer #1:

Remarks to the Author:

In the manuscript by Mari et al., the authors present a detailed characterization of the ability for gasdermin A3 to form diverse oligomers. They use atomic force microscopy to detail size and shapes of arc, slit and ring-like oligomers of the N-terminal domain of gasdermin A3. They then report transmission electron microscopy images of these oligomers formed on liposomes and negatively stained. Finally, they use molecular dynamics simulations to model protein conformational changes of the oligomer and to model how lipids are re-ordered during pore formation. I have several issues with technical aspects of the manuscript as well as the impact of the findings.

Figure 1:

One of the main concerns I have is that the authors use AFM height measurements to define if an oligomer is in a prepore or pore state. While clearly pores are observed, based their static nature and clear depression in the lumen of the channel, I do not think there is sufficient evidence that prepore states are being measured. The authors use the lack of depression in the lumen of the oligomer as evidence of a prepore. These AFM measurements are highly dependent on the sharpness of the tip. If the tip is slightly dull, you won't get the sensitivity to see the depression in between. Also, it is extensively documented that prepores are highly mobile on supported lipid bilayers and not easily measurable by AFM due to being pushed around by the tip. Prepore assemblies often need to be immobilized before measuring, either by changing the lipid phase or increasing the protein density on the supported lipid bilayer as to restrict mobility.

Figure 2:

Here the authors use negative stain TEM to visualize gasdermin pores on liposomes. They clearly show that GSDMA3_{nterm} alone do not make oligomers, and that liposomes either alone, with uncleaved GSDM or with TEV do not make oligomers. However, due to distortions of collapsed liposomes in negative stain, I do not think they can draw any significant conclusions of the shape of gasdermin oligomers from these images. Either negatively stained oligomers on lipid monolayers, or cryoEM images of pores in liposomes would be needed.

Figure 3:

Here the authors use AFM to characterize the variability in gasdermin oligomers. Their data clearly show a variety of stoichiometries and a combination of arc, slit and ring shapes. Again, I respectfully disagree with how they define "prepore" vs pore oligomers in panel E. Other pore forming proteins (perforin and CDCs) have specifically captured conformations using disulphide locked variants. In these cases, locked variants are tested for pore-forming ability using liposome dye leakage assays. The trapped state can then be released with reducing agent, and the pore transition tracked using AFM. While it is highly plausible that similar to perforin and MAC pores, gasdermin pore formation does not require a vertical collapse towards the lipid bilayer, I don't think the authors provide sufficient evidence here to support that conclusion. I also found their supplementary figure (Fig. S3) which supports their time-lapsed AFM very difficult to follow. A movie of a small zoomed in area tracked over the duration of the measurement may be easier to follow single pores. If propagating oligomers are being tracked in real-time by AFM, perhaps another interpretation could be that gasdermin forms growing pores (like perforin and MAC) rather than a prepore to pore transition (as in CDCs)?

Figure 4:

In the final section of the paper, the authors present both coarse grain and all-atom MD simulations of gasdermin pores in lipid bilayers. Based on these simulations, the authors identify flexible regions of the structure that could be important to accommodate the diverse stoichiometries observed in AFM images. These data bring up an exciting structural hypothesis which would have been strengthened by mutagenesis data probing the significance of these amino acids. There is no mention of the properties

of these amino acids or their conservation across the broader gasdermin family of proteins. In addition, the authors describe various trajectories for lipids under different pore stoichiometries. I think this is the most interesting aspect of the simulations and more could have been done to discuss these results within the context of other pore forming proteins. While the Voegelé et al., PNAS paper is cited, these results were not really discussed. In this paper the authors also observe similar lipid re-arrangements including the formation of a vesicle that gets released from the pore lumen.

Discussion section:

I found the discussion section for this paper speculative with several vague statements. More could have been done to put this work in the context of other pore forming proteins. While many of the relevant works are cited, they are lumped together in broad statements rather than discussing how their results compare or what new insight this study provides the field. From my reading, the main finding of this paper is the characterization of oligomeric states of gasdermin pores. Many other pore-forming proteins (both CDCs and perforin) form similar arc, slit and ring like assemblies. It is not clear to me what the functional significance of these various stoichiometries are.

Reviewer #2:

Remarks to the Author:

This manuscript provides details on the mechanism of membrane interactions of gasdermin A3 (GSDMA3), a protein belonging to the gasdermin protein family. This family representatives are main effector proteins of pyroptosis and have thus attracted considerable attention in recent years. Hereby authors studied interaction of the Nterminal fragment of GSDMA3 (GSDMA3Nterm) with lipid membranes. They have used atomic force microscopy and molecular modelling to provide insights into mechanism of membrane damage induced by GSDMA3Nterm.

Some of the insights provided in this manuscript were previously shown by the same approach by the same authors for other members of this family and are thus not novel (i.e. formation of arc-, slit- and ring-shaped oligomers; i.e. pape 21). The novel aspects of the work is extensive modelling of membrane extrusion by different oligomers, however, this is left more or less at the modelling level and no experimental data is provided. Moreover, some findings about flexibility of GSDMA3Nterm molecule and mechanism of pore formation were previously presented at the structural level (i.e. paper by Ruan et al. (2021) Nature vol 557).

Line 42: replace homology with similarity. Homology is a qualitative term.

Line 77: the most recent pore structure of gasdermin A3 should be cited here (Xia et al. (2021) Nature vol 593, 607-611; this reference is listed under #4, but lacks publication details).

Lines 113-115: the lipid composition of the membrane is important, as it can affect the shape of the oligomers. The composition of the lipid extract that was used could be better defined.

Line 119: it is stated that 3h incubation @37deg resulted in formation of GSDMA3Nterm and FigS1B is cited. However, the gel in this figure shows cleavage results after overnight incubation and the temperature is not stated in the legend to Fig. S1B. Supplement the legend with this information and show the data for 3h cleavage in order to better assess the efficiency.

Lines 123-131: Very nice images! Interestingly, membrane insertion is not accompanied with the high change, something that is a hallmark of MACPF/CDC membrane insertion. Could authors comment why some pore profiles in panels Fig1 F and G are of different depth in the membrane? Could some of these indicate partly inserted pores?

Line 136: the height of GSDMA3 oligomers is 2.6 nm. The height of GSDMD is 3.6 nm (based on the Mulvihill 2018). How to explain this difference, since both proteins are structurally very similar?

Line 140: Why are densities of membrane coverage different on images Fig1. A and B, and Fig. 2SA and B? Is this due to different concentration of GSDMA3 used? The concentrations of GSDMA3 in these panels should be specified in legends to corresponding figures, similar as for GSDMA3 and TEV, which is stated in the legend of Fig. S2 in description of panels C and D.

Line 145: Legend to Fig. S3: the white arrowhead is mentioned (line 49), but I cannot see it on the image.

Line 198: The assignment of prepores and pores may be tricky. Similar question as above for Fig. 1.

Line 216: Reference Ruan is not cited as other references. Please correct.

Line 219: Fig. 4E shows comparison between AFM topograph and MD topograph for an oligomer with 30-fold stoichiometry. How oligomers with other different stoichiometries compare with the AFM topographs?

Line 267: Fig. S10- Line 149 in supp information: In the legend to Fig. S10, reference to Wimley is not cited as others. Please correct.

Reviewer #3:

Remarks to the Author:

Mari et al. apply various methods to study the pore-forming N-terminal domain of mouse gasdermin A3 (GSDMA3Nterm). AFM topographs reveal oligomers which self-assemble in arc-, slit-, and ring-shaped formations of pre-pores or transmembrane pores. Pre-pores are defined as oligomers with lipid-plugged apertures. The different structures formed immediately and remained stable over long time scales. These observations were confirmed for liposomes studied with TEM. Certain stoichiometries were more stable (18-36), with 30 being most common. Pore and pre-pore states existed indiscriminately for the three observed conformations, suggesting different stages in the pore-forming mechanism. To investigate further, Mari and colleagues used MD simulations. The simulations agree with the AFM data and further reveal that the tilt of the head domains relative to the β -hairpins optimize conditions, both chemical and structural, for a particular range of stoichiometries. The authors hypothesize that this introduces a size limit for exported biomolecules during lysis. Simulations also showed the process of pre-pore to lytic pore by flushing lipid from the opening by water. Lipids escape into the surrounding lipid, or into the solution as nanodiscs or liposomes.

Overall, the authors present a comprehensive study of the pore-formation mechanism of gasdermin-A3. The results will be of interest to a wide audience and showcase a powerful combination of techniques: AFM, TEM, and MD simulations. The experimental work, including especially the AFM, is of very high quality and the analysis appears solid. The work is well suited for eventual publication in Nat Communications.

Comments/questions:

- Line 155: In what conditions were these samples stored for long-term experiments?
- Line 175: It should be stated how the number of subunits was determined for the oligomers. I'm assuming, since the data is very high resolution, that they could simply be counted the distinct gaussian-like peaks, either from the raw data or the correlation averages.
- Lines 218-9: What metric is used to assess the excellent agreement between the MD topograph and the AFM topograph?
- Lines 383-4: "... and 0.4 μ M TEV at 37 °C in imaging buffer (...) at RT." It's not clear here what temperature was used for this incubation – two temperatures are given.
- Lines 428-9: How was drift determined here? Did you assume linear drift velocity and re-scale? Also, I guess that this drift varied day to day. It would be useful to know what was the range of drift rates observed in the instrument.
- Line 580: "...the liposomes fused fused into membrane patches..." ↓ repeated word
- Line 586: "(M) The mixture..." ↓ panel label format: bold.
- Line 591: "(Methods)" ◇ is this a reference to the methods section?
- There is no methods section for the TEM data. It is described in the Figure 2 caption, but not anywhere else.
- General comment – two distinct modes of AFM imaging are discussed in the text and methods, but it

is not clear which mode was used for each data set presented. To clarify, this information should be included in the captions.

Point-by-Point Response to the Reviewer's Comments

Point-by-Point Response to Reviewer #1

Reviewer #1: In the manuscript by Mari et al., the authors present a detailed characterization of the ability for gasdermin A3 to form diverse oligomers. They use atomic force microscopy to detail size and shapes of arc, slit and ring-like oligomers of the N-terminal domain of gasdermin A3. They then report transmission electron microscopy images of these oligomers formed on liposomes and negatively stained. Finally, they use molecular dynamics simulations to model protein conformational changes of the oligomer and to model how lipids are re-ordered during pore formation. I have several issues with technical aspects of the manuscript as well as the impact of the findings.

Authors: We thank the reviewer for his/her critical and constructive comments, which have guided us to improve our manuscript. Below we answer point-by-point how we addressed each comment of the reviewer.

Reviewer #1: Figure 1: One of the main concerns I have is that the authors use AFM height measurements to define if an oligomer is in a prepore or pore state. While clearly pores are observed, based their static nature and clear depression in the lumen of the channel, I do not think there is sufficient evidence that prepore states are being measured. The authors use the lack of depression in the lumen of the oligomer as evidence of a prepore. These AFM measurements are highly dependent on the sharpness of the tip. If the tip is slightly dull, you won't get the sensitivity to see the depression in between.

Authors: The reviewer acknowledges that we clearly observe oligomers residing in the pore state. However, he/she thinks that there is not sufficient evidence that a pre-pore state is measured. As per definition the oligomers reside in the pre-pore state if the lytic transmembrane pore has not been formed yet (*i.e.*, the pore remains plugged). The reviewer argues that this plugged state may be an 'imaging artifact' based on the sharpness of the tip. It is indeed correct that a blunt or contaminated AFM tip may cause imaging artifacts and pores may be observed as plugged pores¹. However, this cannot occur for open and plugged pores, which are imaged in the same topograph and thus with the same AFM tip. We kindly note that in our AFM topographs we observe murine GSDMA3^{Nterm} (mGSDMA3^{Nterm}) oligomers to co-exist both in the pre-pore state (pore plugged) and pore state (open transmembrane pore). Several of such topographs showing individual oligomers residing in the pre-pore and pore state are presented in our Manuscript (see **Fig. 1A, 1B; Supplementary Fig. S2A, S2B, S4A, S4B, S5A, S5B, S6A, S6B**). As an example, we here show **Fig. R1** (taken from **revised Fig. 1**) and if the reviewer wishes to see more, we can show examples of the height profiles of oligomers residing in either the pre-pore or pore state from all above-mentioned topographs.

Figure R1, included as revised Figure 1. Murine GSDMA3^{Nterm} assembles a variety of oligomeric shapes and sizes, which can adopt either the pre-pore or pore state, in lipid membranes. A and B, AFM topographs of mGSDMA3^{Nterm} oligomers formed on supported lipid membranes (SLMs) made from *E. coli* polar lipid extract. C-E, AFM topographs showing (C) arc-, (D) slit-, and (E) ring-shaped mGSDMA3^{Nterm} oligomers. F and G, Height profiles of mGSDMA3^{Nterm} oligomers measured along the red lines indicated in the AFM topographs (A, B). Numbers in the upper right corner correlate height profiles to red lines in topographs. Black dashed lines represent the membrane surface (0 nm height). Pre-pore and pore states are indicated for each oligomer. Pore state has been defined for oligomers, whose pores protrude ≥ 2 nm into the lipid membrane. The FD-based AFM topographs were recorded in imaging buffer solution and exhibit a full-range color scale corresponding to a vertical scale of 6 nm. Scale bars, 50 nm (A, B) and 20 nm (C-G). H, Diameters of maximum height of arc-, slit- and ring-shaped mGSDMA3^{Nterm} oligomers imaged by AFM in membranes made from *E. coli* polar lipid extract. Black curves represent Gaussian fits determining the mean \pm SD values given. *n* gives the number of oligomers analyzed. I, Maximum heights of arc-, slit- and ring-shaped mGSDMA3^{Nterm} oligomers protruding from the lipid membrane surface and either residing in the pre-pore or pore state. Values present mean \pm SD. Statistics, averages and errors are summarized in Supplementary Table S1 and S2.

We also report another example, which even more convincingly shows the co-existence of oligomers having closed (pre-pore state) or open (pore state) pores (Fig. R2). Whereas the ring-shaped oligomer no. 1 shows clearly an open transmembrane pore that penetrates the lipid membrane (depth ≥ 3 nm, Fig. R2), the pores of the ring-shaped oligomers no. 2 and no. 3 are plugged most probably by the lipid membrane, thus indicating their pre-pore state. The ring-shaped oligomers between no. 1 and no. 2 and the arc-shaped oligomer on the right side of the ring no. 3 also have to be considered pre-pores since their lumen appears plugged by smaller arc-shaped oligomers having the same height above the membrane as the oligomer external walls. In summary, the examples show mGSDMA3^{Nterm} oligomers residing in both pre-pore and pore states in the same AFM topograph. Because the oligomers residing in both states have been imaged using the same AFM tip, we can exclude that mGSDMA3^{Nterm} oligomers showing a plugged pore and oligomers showing an open lytic (transmembrane) pore are a tip artifact.

Figure R2. Additional example of mGSDMA3^{Nterm} oligomers assembled in the lipid membrane and adopting either the pre-pore state or pore state. **A**, FD-based AFM topograph of a densely packed assembly of mGSDMA3^{Nterm} oligomers inserted in a supported lipid membrane made from *E. coli* polar lipid extract. The red line indicates the height profile shown in **(B)**. The sample was prepared and the AFM topograph was recorded as described in Methods. Scale bar, 30 nm. The full-range color scale of the AFM topograph corresponds to a vertical scale of 9 nm. **B**, Height profile extracted from the AFM topograph shown in **(A)**. Numbers 1, 2, and 3 correlate to ring-shaped oligomers along the red height line in **(A)**. The height of 0 nm indicates the surface of the lipid membrane. The ring-shaped oligomer no. 1 shows a depth of ≈ 3.3 nm indicating that the oligomer resides in the pore state. The ring-shaped oligomers no. 2 and 3 are filled with lipids having the same height as the surface of the surrounding lipid membrane. Hence these rings reside in the pre-pore state. Scale bar, 30 nm.

Reviewer #1: *Figure 1: ... continued ...* Also, it is extensively documented that prepores are highly mobile on supported lipid bilayers and not easily measurable by AFM due to being pushed around by the tip. Prepore assemblies often need to be immobilized before measuring, either by changing the lipid phase or increasing the protein density on the supported lipid bilayer as to restrict mobility.

Authors: Upon repetitively imaging the same membrane areas by AFM, the topographs show that mGSDMA3^{Nterm} oligomers, which have inserted into the membrane, are not pushed away by the AFM tip (Fig. R3). In addition, we show mGSDMA3^{Nterm} oligomers residing either in the pre-pore or in the pore state, independently of the protein density (Fig. 1A, 1B, 4A; Supplementary Fig. S2A, S2B, S4A, S4B, S5A, S5B; and Fig. R2) on the supported lipid membrane. We can thus exclude that oligomers residing in the membrane-inserted pre-pore state such as reported in our paper are (i) highly mobile or/and (ii) depend on the protein density.

Figure R3. Repetitive AFM imaging of mGSDMA3^{Nterm} oligomers inserted into supported lipid membranes (SLMs) made from *E. coli* polar lipid extract shows that mGSDMA3^{Nterm} oligomers are not pushed away by the AFM tip. A section of the area imaged in the topograph no. 1 (approximately the area indicated by the white rectangle) has been repetitively imaged for a total of 9 consecutive topographs at different resolutions and magnifications. None of the oligomers contoured has been displaced by the AFM tip. Scale bars, 30 nm. The full-range color scale of the FD-based AFM topographs corresponds to a vertical scale of 5.4 nm. Imaging condition and buffer solution as described for the other FD-based AFM images of the manuscript (**Methods**).

It is worth noting that the AFM topographs displayed in **Fig. R3** just like the other topographs shown in the Manuscript and Supplementary (with the exception of **new Fig. 2**, **revised Supplementary Fig. S3** and **new Supplementary Movie S1**) have been captured after the SLM has been thoroughly rinsed with buffer solution to remove mGSDMA3 and TEV from the solution. On the contrary the AFM time-lapse experiments shown in **Fig. 2**, **Supplementary Fig. S3** and **Movie S1**, were recorded while incubating the membrane with mGSDMA3^{Nterm}. In these cases, in addition to the membrane-inserted oligomers residing in the pre-pore and pore states described before, we also observe mobile ring-shaped oligomers diffusing on the lipid membrane similarly to the ‘pre-pores’ mentioned by the reviewer. The appearance of these oligomers increases as the time-lapse AFM imaging proceeds and mGSDMA3^{Nterm} oligomers gradually assemble and insert into the membrane (hence the protein density on the lipid membrane increases). Under well controlled AFM imaging conditions (*e.g.*, low imaging force and thermal equilibration of the AFM) the mobile oligomers can be contoured by the AFM tip as ring-shaped (**Fig. R4**). Such mobile oligomers, which are not observed to form transmembrane pores, are removed by thoroughly rinsing the membrane with buffer solution (**new Supplementary Fig. S4**). This indicates that the mobile oligomers are only weakly attached (*i.e.*, adsorbed) onto the membrane, possibly interacting with the lipids *via* the α 1 helix and/or the β 1– β 2 loop (**new Supplementary Fig. 12**), such as shown by our MD simulations and previously suggested by others^{2,3}. The mobile membrane-attached ring-shaped oligomers closely resemble the solubilized mGSDMA3 and hGSDMD ring-shaped oligomers without β -barrel such as described by Ruan *et al.*² and Xia *et al.*³.

We hope that the reviewer can now better understand our point of view. We define oligomers as residing in a pre-pore state if they do not form open transmembrane pores. Consequently, membrane-attached oligomers which are (i) mobile and washed away by rinsing with protein-free buffer solution and (ii) not forming transmembrane pores are considered to reside in a pre-pore state. Membrane-inserted oligomers that do not form an open transmembrane pore also reside in a pre-pore state, while membrane-inserted oligomers that form an open transmembrane pore, are considered to reside in the pore state. It is also important to mention that (i) membrane-attached mGSDMA3^{Nterm} oligomers, (ii) transmembrane mGSDMA3^{Nterm} oligomers residing in the pre-pore state (i.e., plugged pores), and (iii) transmembrane mGSDMA3^{Nterm} oligomers residing in the pore state (i.e., transmembrane lytic pores) all show the same height above the membrane. This demonstrates that neither the oligomer insertion nor the pore formation requires a vertical (e.g., height above the membrane) collapse (**Fig. R1I** and **Fig. R4J**).

Figure R4, now included as new Fig. 2 into the revised Manuscript. Time-lapse AFM sequences showing mobile membrane-attached and immobile membrane-inserted mGSDMA3^{Nterm} oligomers. A–H, a defect-free SLM made from *E. coli* polar lipid extract was incubated with imaging buffer solution containing 1.5 μ M mGSDMA3, which had been beforehand cleaved with 0.4 μ M TEV overnight at 37°C, and imaged in the same solution at 37°C. Recorded at different time points of the incubation (time stamps indicate minutes), the time-lapse AFM topographs monitor the assembly, disassembly and diffusion of mGSDMA3^{Nterm} oligomers. From A to H, the topographs follow over the time course different areas of the SLM. The central topographs capture mobile ring-shaped mGSDMA3^{Nterm} oligomers (indicated by white arrows) which were not there previously (topograph on the left side) and which thereafter disassemble or change position (topograph on the right side). The time-lapse FD-based AFM topographs were recorded in imaging buffer solution at 37°C as described (Methods**), and their full-range color scale corresponds to a vertical scale of 6 nm. Scale bar of 50 nm applies to all topographs. I, Height profiles of mobile mGSDMA3^{Nterm} oligomers measured along the red lines indicated in the AFM topographs (D, H). Numbers in the upper right corner correlate height profiles to red lines in topographs. Black dashed lines represent the membrane**

surface (0 nm height). **J**, Maximum heights of ring-shaped oligomers residing in the membrane-attached (MA) and membrane-inserted pre-pore (MI pre) and pore (MI pore) state. Values present mean \pm SD. Statistics, averages and errors are summarized in **Supplementary Table S1**. **K**, Diameters of the maximum height of ring-shaped mGSDMA3^{Nterm} oligomers residing in the membrane-attached (MA) and membrane-inserted pre-pore (MI pre-pore) and pore (MI pore) state and imaged by time-lapse AFM. Black curves represent Gaussian fits determining the mean \pm SD values given. *n* gives the number of oligomers analyzed. Statistics, averages and errors are summarized in **Supplementary Table S2**. For further information on the representative time-lapse AFM series see **Supplementary Fig. S3** and **Movie S1**.

In summary, we apologize that in our initially submitted manuscript we did not address the existence and properties of the mobile membrane-attached mGSDMA3^{Nterm} oligomers at sufficient clarity and detail. The reason for this was that we initially thought to focus our paper on the high structural flexibility of the membrane-inserted mGSDMA3 oligomers and on how these oligomers force the lipids to exit the pore lumen. However, encouraged by the reviewer's comments, we now expanded our description. Namely, we observe:

1. Membrane-attached mGSDMA3^{Nterm} oligomers, which are weakly attached to the membrane surface, mobile and not forming transmembrane pores. Because these membrane-attached oligomers do not form an open transmembrane pore we describe them to reside in a pre-pore state.
2. Membrane-attached mGSDMA3^{Nterm} oligomers, which are inserted into the membrane but plugged with lipids and/or proteins and therefore not yet forming transmembrane pores. We describe these membrane-inserted oligomers to also reside in a pre-pore state.
3. Membrane-inserted mGSDMA3^{Nterm} oligomers forming open (lytic) transmembrane pores. We describe these oligomers to reside in the pore state.

In the revised version of the manuscript we describe and discuss these differences and states more clearly. To do so we also show the **new Fig. 2** (here shown as **Fig. R4**), **revised Supplementary Fig. S3** and **new Supplementary Movie S1**, include controls washing away the mobile membrane-attached oligomers (**new Supplementary Fig. S4**), and include a conclusive figure (**new Fig. 7**) (see **revised Manuscript**, sections Abstract, Results, Discussion, and Supplementary).

Reviewer #1: Figure 2: Here the authors use negative stain TEM to visualize gasdermin pores on liposomes. They clearly show that GSDMA3^{Nterm} alone do not make oligomers, and that liposomes either alone, with uncleaved GSDM or with TEV do not make oligomers. However, due to distortions of collapsed liposomes in negative stain, I do not think they can draw any significant conclusions of the shape of gasdermin oligomers from these images. Either negatively stained oligomers on lipid monolayers, or cryoEM images of pores in liposomes would be needed.

Authors: The reviewer finds our negative stain TEM experiments to convincingly show that mGSDMA3^{Nterm} requires lipid membranes to assemble oligomers. However, the reviewer questions whether collapsed liposomes imaged by negative stain TEM allow to draw conclusions on the oligomeric shape. We would like to draw the kind attention of the reviewer to the fact that also flat and most likely not collapsed lipid membranes like those shown in **Fig. 3D, 3G, 3I** show the coexistence of arc-, slit- and ring-shaped oligomers. Besides our present and previous TEM work on hGSDMD⁴, also other authors have published TEM data of gasdermins in different lipid compositions showing oligomers of different shapes⁵. These oligomers have circular shapes and shapes significantly deviating from a ring-shape (**Fig. R5**). Ding *et al.*⁵ also show negatively stained hGSDMD^{Nterm} and mGSDMA3^{Nterm} oligomers on lipid

monolayers composed of 80% phosphatidylcholine and 20% cardiolipin clearly having different shapes and sizes (Fig. R5).

Ding, J., Wang, K., Liu, W. et al. Pore-forming activity and structural autoinhibition of the gasdermin family. *Nature* 535, 111–116 (2016)

Adapted from Figure 3 | Membrane pore-forming activity of the gasdermin-N domain.

a, b, Liposomes with indicated lipid compositions (a) or prepared using bovine liver-derived polar lipid extracts (b) were treated with indicated gasdermin proteins. Shown are representative negative-stain electron microscopy micrographs of the liposomes (scale bar, 100 nm). Insets in a, expanded view of a representative pore (scale bar, 15 nm). All data shown are representative of three independent experiments.

Adapted from Extended Data Figure 6

Pores formed by active GSDMD and GSDMA3 on monolayer membranes containing 80% phosphatidylcholine and 20% cardiolipin. Shown are representative negative-stain electron microscopy micrograph images (scale bar, 100 nm).

Figure R5, adapted from Ding et al. 'Pore-forming activity and structural autoinhibition of the gasdermin family' *Nature* 535, 111-116 (2016). Top panel adapted from Fig. 3 shows representative TEM images of negatively stained hGSDMD and mGSDMA3 oligomers on liposomes with the indicated lipid compositions (a) or prepared using bovine liver-derived polar lipids extracts (b). Bottom panel adapted from Extended Data Fig. 6 of the paper Ding et al., shows representative TEM images of negatively stained hGSDMD and mGSDMA3 oligomers on

monolayer membranes containing 80% phosphatidylcholine and 20% cardiolipin. All TEM images show gasdermin pores having both circular and irregular shapes.

As for the cryo-TEM imaging of gasdermin oligomers in liposomes suggested by the reviewer, Sborgi et al.,⁶ used cryo-TEM to visualize hGSDMD pores in liposomes (**Fig. R6**). In these preparations, the liposome surface features numerous oligomers formed by hGSDMD^{Nterm}. Although the liposomes appear to be completely covered by oligomers, from the available contrast of the cryo-TEM micrographs, it remains inconclusive, whether the inside of the oligomers is lipid-filled (pre-pore state) or actually devoid of lipid (pore state). The lack of contrast also hardly allows to clearly identify the various shapes of the hGSDMD oligomers. In order to understand whether the oligomers reside in the lipid-filled pre-pore state or in the open lytic transmembrane pore state and to identify the different shapes of the oligomers, the authors used AFM, which enables to contour the surface of native proteins at sub-nanometer resolution⁷. The best attainable lateral and vertical resolutions of single membrane-inserted proteins approach indeed 0.5–0.7 nm and ~0.1 nm, respectively⁸⁻¹⁰, at a signal-to-noise ratio that is considerably higher than attainable by cryo-TEM on individual membrane proteins (without averaging).

Figure R6, adapted from Sborgi et al.,⁶. Visualization of hGSDMD^{Nterm} pores in liposomes by cryo-TEM. A–C, Cryo-TEM micrographs of hGSDMD^{Nterm} pores in *E. coli* polar lipid liposomes. The micrographs were acquired at protein/lipid molar ratios of 1/1,000, 1/500, and 1/100, respectively. Black arrows indicate ring-shaped structures corresponding to oligomeric hGSDMD^{Nterm}. Scale bars, 80 nm. D, Proteoliposome with protein/lipid molar ratio of 1/100 at higher magnification. Black arrows indicate ring-shaped hGSDMD^{Nterm} oligomers. Scale bars, 80 nm.

However, as the reviewer finds that the distortions of collapsed liposomes could prevent drawing significant conclusions on the shapes of gasdermin oligomers, we have recorded negative stain TEM images of mGSDMA3^{Nterm} oligomers searching specifically for flat, single layered lipid membranes. The representative TEM image of negatively stained mGSDMA3^{Nterm} pores in a flat and large lipid membrane clearly shows oligomers of different sizes and shapes to co-exist (**Fig. R7**). We kindly remind that in addition to ring-shaped oligomers, arc-shaped oligomers have also been observed for other membrane pore forming proteins (toxins) like perfringolysin O¹¹ and that arcs seem to share the ability to form active pores by excluding lipids, like those formed by hGSDMD^{4,6}, streptolysin O¹², suilysin¹³, Bax¹⁴, perforin¹⁵, listeriolysin O¹⁶ and pneumolysin¹⁷.

Figure R7. Transmission electron microscopy (TEM) of negative stained mGSDMA3^{Nterm} oligomers inserted into single layered lipid membranes made from *E. coli* polar lipid extract. TEM image of liposomes after incubation with mGSDMA3 in the presence of TEV. Upon adsorption onto TEM grids the liposomes opened and fused into single-layered membrane patches that spread out over the carbon film of the grid. mGSDMA3^{Nterm} oligomers inserted into lipid membranes appear to have different shapes and sizes. Liposomes made from *E. coli* polar lipid extract were incubated overnight at 37°C with 7 μM mGSDMA3 and 1.5 μM TEV.

Reviewer #1: Figure 3: Here the authors use AFM to characterize the variability in gasdermin oligomers. Their data clearly show a variety of stoichiometries and a combination of arc, slit and ring shapes. Again, I respectfully disagree with how they define “prepore” vs pore oligomers in panel E. Other pore forming proteins (perforin and CDCs) have specifically captured conformations using disulphide locked variants. In these cases, locked variants are tested for pore-forming ability using liposome dye leakage assays. The trapped state can then be released with reducing agent, and the pore transition tracked using AFM.

Authors: As addressed in answering the earlier question on the same topic, pre-pore *versus* pore state, we hope that the reviewer now can better see our point and possibly agree on how we define the three kinds of mGSDMA3^{Nterm} oligomers we observe:

1. Membrane-attached oligomers (**Fig. R4**) are mobile and do not form transmembrane pores. These oligomers can be easily washed away by rinsing with buffer solution and can be considered pre-pores.
2. Membrane-inserted oligomers residing in the plugged pre-pore state.
3. Membrane-inserted oligomers residing in the open (lytic) transmembrane pore state.

Below we show an additional example of membrane-inserted mGSDMA3^{Nterm} oligomers residing in the pre-pore state (transmembrane oligomer with plugged pore) and in the pore state (transmembrane oligomer with open pore) imaged with the same AFM tip and in the same topograph (**Fig. R8**). The pore of oligomer no. 1 of **Fig. R8A** penetrates the membrane by almost 5 nm and therefore is undoubtedly an open lytic transmembrane pore. The

oligomer furthermore has the same height above the membrane (≈ 3 nm) as oligomer no. 2, whose pore is plugged and therefore resides in the pre-pore state. In this example we also capture the transition of the mGSDMA3^{Nterm} oligomer no. 2 from the pre-pore state (Fig. R8A) to the pore state (Fig. R8B). Independent of whether the mGSDMA3^{Nterm} oligomer no. 2 resides in the pre-pore (Fig. R8A) or pore (Fig. R8B) state it protrudes by the same height above the membrane (≈ 3 nm), thus showing that the transition from the pre-pore to the pore state occurs in the absence of vertical collapse. We have included Fig. R8 into our revised Manuscript as new Supplementary Fig. S6.

Figure R8, included in the revised Manuscript as new Supplementary Fig. S6. Imaging a mGSDMA3^{Nterm} oligomer transiting from the pre-pore to the pore state in the absence of vertical collapse in supported lipid membranes made from *E. coli* polar lipid extract. **A and B**, FD-based AFM topographs recorded after each other show a mGSDMA3^{Nterm} oligomer transiting from the pre-pore (plugged) to a pore (unplugged) state. The red lines along the ring-shaped oligomers indicate the height profiles shown in **C** and **D**. Scale bars, 30 nm. The full-range color scale of the AFM topographs corresponds to a vertical scale of 10 nm. Imaging condition and buffer solution as described for the other AFM images (Methods). **C and D**, Height profiles showing the transition of mGSDMA3^{Nterm} oligomer no. 2 from the plugged, pre-pore state in **C** to the unplugged, pore state in **D**, whereas mGSDMA3^{Nterm} oligomer no. 1 remains in the unplugged, pore state. Independent of whether they reside in the pre-pore or pore state the oligomers show the same heights above the membrane.

Reviewer #1: Figure 3: ... While it is highly plausible that similar to perforin and MAC pores, gasdermin pore formation does not require a vertical collapse towards the lipid bilayer, I don't think the authors provide sufficient evidence here to support that conclusion.

Authors: The evidence that supports our conclusion that mGSDMA3, similarly to hGSDMD, does not undergo a vertical collapse to form lytic, transmembrane pores is:

1. Mobile membrane-attached and membrane-inserted mGSDMA3^{Nterm} oligomers residing in the pre-pore state or pore state all protrude at the same height above the membrane (Fig. R2, R8, Fig. 1, 4, 6, revised Supplementary Table S1).

2. We could capture membrane-inserted mGSDMA3^{Nterm} oligomers transiting from the plugged pre-pore state to the open pore state (**new Supplementary Fig. S6**). The oligomers do not show height changes and thus do not undergo a vertical collapse.
3. All mGSDMA3^{Nterm} oligomers observed in our time-lapse AFM experiments (while the lipid membrane is being incubated with mGSDMA3^{Nterm}, which progressively inserts and assembles oligomers in the membrane) have the same height above the membrane (**new Fig. 2, revised Supplementary Fig. S3, new Supplementary Movie S1, and revised Supplementary Table S1, S2**).
4. Simulations of arc-, slit-, and ring-shaped mGSDMA3^{Nterm} oligomers transiting from the pre-pore (plugged pore) to the pore (open pore) state show that when the hydrophilic inner face of the β -sheet or of the β -barrel draws water into the protein–lipid interface, forcing lipids to recede and unplug the pore, the oligomers do not change height (**revised Fig. 6, Supplementary Fig. S15**).

It is also worth to mention that evidence of the absence of vertical collapse also stems from the cryo-TEM structures of both mGSDMA3^{Nterm} and hGSDMD^{Nterm} ring-shaped oligomers that show the globular (head) domain to have comparable heights in the absence and presence of a β -barrel^{2,3}. This is a strong indication that the globular head domain does not change height when the β -barrel domain integrates into the membrane and supports our data showing the absence of a vertical collapse. We have revised our Manuscript to address this issue more clearly (see **revised Abstract, Results and Discussion**)

Reviewer #1: I also found their supplementary figure (Fig. S3) which supports their time-lapsed AFM very difficult to follow. A movie of a small zoomed in area tracked over the duration of the measurement may be easier to follow single pores.

Authors: In our time-lapse AFM experiments we observe a lively scenario of oligomers diffusing, inserting, growing, and changing shapes. Although we find it fascinating to observe these processes in our time-lapse AFM images in full screen, we agree with the reviewer that it is difficult to follow the many details in the former **Supplementary Fig. S3**. As suggested by the reviewer, we have **revised Supplementary Fig. S3** to show the oligomeric assembly process in more detail, show small enlarged areas of the time-lapse AFM experiments in more detail in new Figures (**Fig. R4, now included as new Fig. 2, and revised Supplementary Fig. S3**), and show the time-lapse AFM in a movie (**new Supplementary Movie S1**). Indeed both, the new figure and movie, make it easier to follow the diffusion, assembly and growth of single membrane-inserted oligomers and the assembly and diffusion of membrane-attached oligomers (see **revised Manuscript**).

Reviewer #1: If propagating oligomers are being tracked in real-time by AFM, perhaps another interpretation could be that gasdermin forms growing pores (like perforin and MAC) rather than a prepore to pore transition (as in CDCs)?

Authors: We agree with the reviewer that the AFM data suggest a model of gasdermin growing oligomers. We indeed observe smaller membrane-inserted slit- and arc-shaped oligomers to gradually grow into larger oligomers, which indicates a “membrane-inserted oligomerization” (*e.g.*, the oligomers assemble and grow in the membrane). An alternative model would be the “membrane-attached oligomerization”, in which the oligomers first assemble on the membrane surface and in a second step insert already assembled oligomers into the membrane. Interestingly, we also observe the mobile membrane-attached ring-shaped oligomers (**Fig. R4, included as new Fig. 2**) as they would occur in the “membrane-attached oligomerization” model. Hence, our experimental data suggests that both the

“membrane-inserted oligomerization” and the “membrane-attached oligomerization” pathways coexist. This said, we have to keep in mind that the buffer composition, the lipid membrane composition, etc. may play a role and favor the “membrane-inserted oligomerization” compared to the “membrane-attached oligomerization” or *vice versa*. Accordingly, to address the comment raised by the reviewer, we have **revised the Results and Discussion sections** of our Manuscript.

Reviewer #1: Figure 4: In the final section of the paper, the authors present both coarse grain and all-atom MD simulations of gasdermin pores in lipid bilayers. Based on these simulations, the authors identify flexible regions of the structure that could be important to accommodate the diverse stoichiometries observed in AFM images. These data bring up an exciting structural hypothesis which would have been strengthened by mutagenesis data probing the significance of these amino acids. There is no mention of the properties of these amino acids or their conservation across the broader gasdermin family of proteins. In addition, the authors describe various trajectories for lipids under different pore stoichiometries. I think this is the most interesting aspect of the simulations and more could have been done to discuss these results within the context of other pore forming proteins. While the Vogele et al., PNAS paper is cited, these results were not really discussed. In this paper the authors also observe similar lipid re-arrangements including the formation of a vesicle that gets released from the pore lumen.

Authors: Thank you. It appears that in our attempt to keep the manuscript compact and focused, we have sacrificed some relevant information. We now mention the properties of the amino acids of the flexible structural regions of mGSDMA3^{Nterm} and analyze their conservation across the gasdermin family (see **Table R1** at the end of our response and **new Supplementary Fig. S13**). Briefly, while the β -strand residue V115 assures for the continuous connection of the β -strand of the transmembrane β -hairpin 1 and hydrophilic head domain, the three connector loops 1, 2, and 3 break the continuation of the remaining β -strands to the head domain. In detail, connector loop 1 contains a secondary structure breaking G (G77) and the connector loops 2 and 3 are comprised by "two-too-many" amino acids causing bulging of the loops and bestowing the structure a great flexibility by the possibility to stretch. It is interesting to analyze the conservation of these residues within the gasdermin family members: First, only 2 out of 16 available unique gasdermin sequences do not include a secondary structure breaking G or P residue in the connector loop 1. However, in these two sequences of hGSDMB and mGSDME additional amino acids precede loop 1, thus giving rise to a bulging loop, similar to the loops 2 and 3 described above. The connector β -strand residue V115 is also highly conserved within the gasdermin family (11 out of 16 sequences include a V in this position, one I, one F, and three K). The connector loop 2 consists of the charged or polar residues K161-Q162-E163. In 10 out of 16 sequences a pair of charged amino acids (KE or KD) is found and the second or third position is always either a charged E or a polar Q or N. The connector loop 3, formed by P199-K200-G201, is also very conserved in terms of secondary structure breaking P and G residues (14 out of 16 sequences contain P in the first position, and 12 out of 16 sequences contain G (and one P) in the third position).

The reviewer also suggests the results of our simulations to be discussed in more depth within the context of other pore forming proteins and that the analogies with the results of Vogele *et al.*¹⁸ on the membrane perforation by the toxin pneumolysin should be described. We have revised our Manuscript to better describe our results in the context of other pore forming proteins and especially mentioning the analogies with the results of Vogele *et al.*¹⁸ (see **revised Manuscript**, section Discussion).

Reviewer #1: Discussion section: I found the discussion section for this paper speculative with several vague statements. More could have been done to put this work in the context of other pore forming proteins. While many of the relevant works are cited, they are lumped together in broad statements rather than discussing how their results compare or what new insight this study provides the field. From my reading, the main finding of this paper is the characterization of oligomeric states of gasdermin pores. Many other pore-forming proteins (both CDCs and perforin) form similar arc, slit and ring like assemblies. It is not clear to me what the functional significance of these various stoichiometries are.

Authors: We have revised the discussion and clearly highlighted speculations as such and put our work in the context of other pore forming proteins (see **revised Manuscript**, section Discussion).

The reviewer also asks about the functional significance of the various mGSDMA3^{Nterm} stoichiometries. We have revised our Manuscript to more clearly state our hypothesis that the ability of mGSDMA3^{Nterm} to assemble a wide variety of oligomeric shapes and sizes, each able to form lytic pores may have the function to allow mGSDMA3^{Nterm} to efficiently lyse cell membranes under a variety of conditions. Particularly, the ability to form arc-, slit- and ring-shaped oligomers of different stoichiometries may be key to insert lytic pores in densely packed cellular membranes (see **revised Discussion**).

Point-by-Point Response to Reviewer #2

Reviewer #2: This manuscript provides details on the mechanism of membrane interactions of gasdermin A3 (GSDMA3), a protein belonging to the gasdermin protein family. This family representatives are main effector proteins of pyroptosis and have thus attracted considerable attention in recent years. Hereby authors studied interaction of the Nterminal fragment of GSDMA3 (GSDMA3Nterm) with lipid membranes. They have used atomic force microscopy and molecular modelling to provide insights into mechanism of membrane damage induced by GSDMA3Nterm.

Authors: We thank the reviewer for his/her critical and constructive comments, which have guided us to improve our manuscript. Below we answer point-by-point how we addressed each comment of the reviewer.

Reviewer #2: Some of the insights provided in this manuscript were previously shown by the same approach by the same authors for other members of this family and are thus not novel (i.e. formation of arc-, slit- and ring-shaped oligomers; i.e. pape 21). The novel aspects of the work is extensive modelling of membrane extrusion by different oligomers, however, this is left more or less at the modelling level and no experimental data is provided. Moreover, some findings about flexibility of GSDMA3Nterm molecule and mechanism of pore formation were previously presented at the structural level (i.e. paper by Ruan et al. (2021) Nature vol 557).

Authors: The reviewer writes that some of the insights provided in this manuscript were previously shown by the same approach by the same authors for other members of the gasdermin family and are thus not novel. We kindly agree that some of our observations are similar to what we have previously described for human GSDMD^{Nterm} (hGSDMD^{Nterm}).^{4,6} In this regard, it is correct to mention that our observations contribute to the important finding that different members of the gasdermin family can assemble similarly (at least murine GSDMA3 (mGSDMA3) and hGSDMD). However, such similarities between gasdermin family members are not so obvious taken the fact that their sequence similarity is relatively low. Particularly, members of the gasdermin D family share in average 32% identical and 49% similar residues with the members of the gasdermin A family¹⁹ (see **Table R2** at the end of our response). Additionally, individual gasdermin family members show specific preferences to target certain cell membranes. Moreover, the statement of the referee infers that proteins from the same protein family should not be further characterized because of the expected similarity of their structure and function relationship. Such an argument would *per se* exclude to characterize the structure and function relationship of different gasdermin family members (which so far have been published frequently in Nature family journals) or of members of the GPCR family just to name another example.

In light of the comments of reviewers #1 and #2, we have revised our Manuscript to more clearly outline the novelty of our findings. Briefly, the novel findings we report include:

- 1) mGSDMA3^{Nterm} assembles oligomers, which are weakly attached to the membrane surface and mobile (**Fig. R1, included as new Fig. 2**). These mobile membrane-attached ring-shaped oligomers closely resemble the solubilized ring-shaped oligomers without β -barrel such as solved by cryo-TEM for both mGSDMA3 and hGSDMD^{2,3}. Because these cryo-TEM structures were determined in the absence of a

- membrane, they had been speculated to represent oligomeric forms of gasdermin that precede membrane insertion^{2,3}.
- 2) Inserted into a membrane mGSDMA3^{Nterm} assembles oligomers of a variety of shapes and sizes. We describe these relatively immobile transmembrane oligomers as membrane-inserted oligomers.
 - 3) Membrane-inserted mGSDMA3^{Nterm} oligomers can exist in either a plugged pre-pore state or an open (lytic) pore state independently of their shape and stoichiometry.
 - 4) mGSDMA3^{Nterm} oligomers in the mobile membrane-attached state, in the membrane-inserted pre-pore state, and in the membrane-inserted pore state all show the same height above the membrane. This demonstrates that neither the oligomer insertion nor the pore formation requires a vertical (*e.g.*, height above the membrane) collapse.
 - 5) Distinct structural regions of mGSDMA3^{Nterm} such as the hydrophilic head domain take certain hitherto unknown roles in stabilizing the different membrane-inserted oligomeric shapes and sizes. The exceptional structural flexibility of the head domains is limited, which constrains the possible shapes and sizes GSDMA3^{Nterm} oligomers can adopt.
 - 6) Membrane-inserted mGSDMA3^{Nterm} oligomers transit from the pre-pore to the pore state because the hydrophilic inner surface of the membrane spanning β -sheet or β -barrel draws water into the protein-lipid interface, which forces the lipids to recede. We observe several pathways by which the lipids can leave the pore. The MD simulations also confirm that there is no vertical collapse involved in such opening of the transmembrane pore.
 - 7) Our time-lapse AFM experiments show the membrane supported oligomerization of membrane-attached mGSDMA3^{Nterm} oligomers. However, the data also provides evidence of membrane-inserted oligomerization, which is initiated by mGSDMA3^{Nterm} inserted into lipid membranes and features a continuous growing process of membrane inserted oligomers without pre-pore to pore transitions.²⁰ Taken together the AFM data thus suggests that both the membrane-attached and the membrane-inserted oligomerization pathways co-exist for gasdermins.

Observation no. 1, is supposedly suggested by cryo-TEM for mGSDMA3^{Nterm} and hGSDMD oligomers in the absence of a lipid membrane^{2,3}. However, these ring-shaped oligomers have been produced by reconstitution and posterior solubilization with detergent. It has never been shown that such solubilized gasdermin oligomers assemble and remain attached at the membrane surface where they diffuse freely. Observations no. 2, 3 and partially 4 have been described for hGSDMD but not for any other gasdermin family member. Observations no. 4, 5, 6 and 7 have never been reported for any gasdermin family members yet. By revising our manuscript, we have more clearly described the substantial degree of novelty and advancement of our findings.

Figure R1, included as new Fig. 2 into the revised Manuscript. Time-lapse AFM sequences showing mobile membrane-attached and immobile membrane-inserted mGSDMA3^{Nterm} oligomers. A–H, a defect-free SLM made from *E. coli* polar lipid extract was incubated with imaging buffer solution containing 1.5 μ M mGSDMA3, which had been beforehand cleaved with 0.4 μ M TEV overnight at 37°C, and imaged in the same solution at 37°C. Recorded at different time points of the incubation (time stamps indicate minutes), the time-lapse AFM topographs monitor the assembly, disassembly and diffusion of mGSDMA3^{Nterm} oligomers. From A to H, the topographs follow over the time course different areas of the SLM. The central topographs capture mobile ring-shaped mGSDMA3^{Nterm} oligomers (indicated by white arrows) which were not there previously (topograph on the left side) and which thereafter disassemble or change position (topograph on the right side). The time-lapse FD-based AFM topographs were recorded in imaging buffer solution at 37°C as described (Methods), and their full-range color scale corresponds to a vertical scale of 6 nm. Scale bar of 50 nm applies to all topographs. I, Height profiles of mobile mGSDMA3^{Nterm} oligomers measured along the red lines indicated in the AFM topographs (D, H). Numbers in the upper right corner correlate height profiles to red lines in topographs. Black dashed lines represent the membrane surface (0 nm height). J, Maximum heights of ring-shaped oligomers residing in the membrane-attached (MA) and membrane-inserted pre-pore (MI pre) and pore (MI pore) state. Values present mean \pm SD. Statistics, averages and errors are summarized in Supplementary Table S1. K, Diameters of the maximum height of ring-shaped mGSDMA3^{Nterm} oligomers residing in the membrane-attached (MA) and membrane-inserted pre-pore (MI pre-pore) and pore (MI pore) state and imaged by time-lapse AFM. Black curves represent Gaussian fits determining the mean \pm SD values given. *n* gives the number of oligomers analyzed. Statistics, averages and errors are summarized in Supplementary Table S2. For further information on the representative time-lapse AFM series see Supplementary Fig. S3 and Movie S1.

The reviewer also comments that some findings about the structural flexibility of mGSDMA3^{Nterm} and mechanism of oligomeric pore formation were previously presented at the structural level (*i.e.*, by Ruan *et al.* (2018) Nature vol 557). As our findings build up on existing knowledge, the most important publications have been cited. By comparing the structure of the mGSDMA3^{Nterm} domain in the auto-inhibited, uncleaved mGSDMA3 and in the transmembrane mGSDMA3^{Nterm} oligomer, Ruan *et al.* (2018) described the conformational changes (and therefore the flexibility) of the mGSDMA3^{Nterm} upon membrane insertion to form

long membrane spanning β -strands. In contrast, however, in our manuscript we describe the conformational changes mGSDMA3^{Nterm} undergoes to stabilize oligomers of different shapes and size. Our findings are thus quite different from the previously published work as we do not describe how mGSDMA3^{Nterm} forms long membrane spanning β -strands, but describe how mGSDMA3^{Nterm} adapts structurally to stabilize different oligomeric shapes and sizes. We have revised our Manuscript to more clearly outline the relevance of our work and the novelty of our findings (see **revised Abstract, Introduction, Results and Discussion**). We hope that the reviewer can now see better the significance and novelty of our findings and how they contribute to the understanding of gasdermin attachment, oligomerization, membrane-insertion, and pore formation.

Reviewer #2: Line 42: replace homology with similarity. Homology is a qualitative term.

Authors: Thank you. Homology has been replaced with similarity.

Reviewer #2: Line 77: the most recent pore structure of gasdermin A3 should be cited here (Xia et al. (2021) Nature vol 593, 607-611; this reference is listed under #4, but lacks publication details).

Authors: Thank you. We now cite the reference with completed the publication details.

Reviewer #2: Lines 113-115: the lipid composition of the membrane is important, as it can affect the shape of the oligomers. The composition of the lipid extract that was used could be better defined.

Authors: Thank you. The composition of the *E. coli* polar lipid extract, which is phosphatidylethanolamine:phosphatidylglycerol:cardiolipin, 67:23.2:9.8 wt/wt%, is now reported in the revised Manuscript (see **revised Methods**, section '*Liposome Preparation*').

Reviewer #2: Line 119: it is stated that 3h incubation @37deg resulted in formation of GSDMA3^{Nterm} and FigS1B is cited. However, the gel in this figure shows cleavage results after overnight incubation and the temperature is not stated in the legend to Fig. S1B. Supplement the legend with this information and show the data for 3h cleavage in order to better assess the efficiency.

Authors: We thank the reviewer for her/his observation. The temperature of the overnight incubation is given in the legend of **Supplementary Fig. S1B**: "The digestion was performed over night at 37°C". However, the reviewer has helped us find some inaccuracies in our manuscript. Namely, when performing AFM experiments to investigate mGSDMA3^{Nterm} insertion and pore formation, we applied either one of the two following experimental conditions:

- 1) Incubating the membrane for 3 h at 37°C with full-length mGSDMA3 and TEV.
- 2) Incubating the membrane for 3 h at 37°C (and up to 5–6 h in the case of time lapse AFM experiments) with a solution of mGSDMA3, which beforehand had been cleaved by TEV overnight at 37°C.

Condition no. 1 resulted in a rather low binding of mGSDMA3^{Nterm} to the lipid membrane, which was often insufficient for high-resolution and time-lapse AFM experiments. However, by applying condition no. 2 we obtained considerably higher binding of mGSDMA3^{Nterm} to the lipid membrane, which was more suitable for our needs. Condition no. 2 most likely produces more mGSDMA3^{Nterm} by compensating the low efficiency of the TEV cleavage with a longer

cleavage time. Prompted by the reviewer's observation, we checked how each experiment shown in the Manuscript was conducted and realized that all experiments shown were done using a solution of mGSDMA3, which beforehand had been cleaved by TEV overnight at 37°C (condition no. 2). We apologize for the inaccuracy and have now corrected the information about how mGSDMA3 was pre-cleaved and the membranes incubated in the revised Manuscript. The condition applied to pre-cleave mGSDMA3 is the same we show in **Supplementary Fig. S1B**. To make sure that a prolonged incubation at 37°C would not cause protein degradation, we tested mGSDMA3 stability over time (**Fig. R2**). The SDS-PAGE analysis shows that incubating full-length mGSDMA3 at 37°C for up to 48 h does not cause protein degradation.

Figure R2. Testing mGSDMA3 stability over time. The indicated amounts (μg) of full-length mGSDMA3 were incubated in buffer solution at 37°C for 0, 16, 24 and 48 hours (h) before being analyzed by SDS-PAGE. mGSDMA3 full-length runs like a 53 kDa band and no trace of protein degradation is visible up to 48 h of incubation. Molecular weight markers are annotated in kDa.

The reviewer further mentions the cleavage efficiency. The cleavage efficiency is rather low. **Fig. R3** shows (please note that the left side of the gel is the SDS-PAGE we show in **Supplementary Fig. 1B**) that when incubating overnight at 37°C equivalent amounts of mGSDMA3 and TEV, TEV can roughly cleave half of mGSDMA3 (left part of **Fig. R3**). The cleavage efficiency is similar at 30°C and room temperature (not shown). The right part of **Fig. R3** shows the SDS-PAGE analysis of the digestion products upon cleaving mGSDMA3 by TEV in the same ratio we use in the AFM experiments. In this case, most of mGSDMA3 remains uncleaved but the cleaved portion of mGSDMA3^{Nterm} is enough to observe mGSDMA3^{Nterm} oligomers by AFM. For the AFM experiments we could not increase the digestion product (mGSDMA3^{Nterm}) by increasing the concentration of TEV (as we did for the gel on the left part of **Fig. R3**) otherwise the enzyme would aggregate on top of the inserted gasdermin oligomers disturbing the AFM imaging and contaminating the AFM tip (**Fig. R4**). For the AFM experiments the TEV concentration had to be kept below the experimentally determined threshold of 0.37 μM to prevent the occurrence of such aggregates.

Figure R3. SDS-PAGE analysis of the enzymatic digestion of mGSDMA3 by TEV. The digestion was performed over night at 37°C. The amount of mGSDMA3 and TEV are indicated in µg on top of each column. Cleavage of mGSDMA3 full-length (53 kDa) results in two fragments of 28 kDa (mGSDMA3^{Nterm}) and 25 kDa (mGSDMA3^{Cterm}). Molecular weight markers are annotated in kDa. Note that the TEV sample contains trace impurities, which do not affect the cleavage reaction. Please note that the left side of the SDS-PAGE analysis is shown in **Supplementary Fig. 1B**.

Figure R4. TEV can form aggregates on top of inserted mGSDMA3^{Nterm} oligomers. A defect-free SLM made from *E. coli* polar lipids was incubated with 1.5 µM mGSDMA3 and 0.75 µM TEV in buffer solution at 37°C. The two representative FD-based AFM topographs show the presence of aggregates (bright material) on top of mGSDMA3^{Nterm} oligomers inserted into the SLM. The formation of such aggregates can be prevented using TEV concentrations lower than 0.4 µM. FD-based AFM topographs were recorded in imaging buffer solution at 37°C as described (**Methods**). The full-range color scale of the topographs corresponds to a vertical scale of 13.9 nm. Scale bars are indicated for both topographs.

Reviewer #2: Lines 123-131: Very nice images! Interestingly, membrane insertion is not accompanied with the high change, something that is a hallmark of MACPF/CDC membrane insertion. Could authors comment why some pore profiles in panels Fig1 F and G are of different depth in the membrane? Could some of these indicate partly inserted pores?

Authors: Thank you. We indeed observe that the transition of membrane-inserted mGSDMA3^{Nterm} oligomers from the pre-pore state (plugged pore) to the pore state (open transmembrane pore) is not associated with a height change of the oligomer protruding from the membrane. This transition from the pre-pore to the pore state must thus happen in the absence of a vertical collapse in contrast to what has been previously reported for example for perfringolysin O¹¹, pneumolysin^{18,21} and sullysin¹³ membrane insertion but in agreement to what has been reported for hGSDMD^{4,6} and listeriolysin O¹⁶. To address this issue in better detail we have **revised our Manuscript** (see **revised Results and Discussion**).

The reviewer also asks why some pore profiles in panels **Fig. 1F, 1G** are of different depth in the membrane. The thickness of the supported lipid membrane corresponds to 5.0 ± 0.3 nm (mean \pm SD, $n=33$). AFM measures this thickness as the height of the lipid membrane protruding above the supporting mica. Following our criteria set upon introducing **Fig. 1** and upon defining the pore state, we assume that a membrane which reduces its thickness by roughly its half is not functional anymore. We thus decided to assume that oligomers having reduced the thickness of the membrane (lipids and sometimes also proteins) inside the pore by more than 2 nm have formed a transmembrane pore. The examples of mGSDMA3^{Nterm} oligomers shown in **Fig. 1** and **4**, sometimes show the pore to be fully transmembrane (fully devoid of lipids or proteins). In other cases, lipids or proteins may still be inside the lumen of the oligomer (see high-resolution AFM topographs of the Manuscript, *e.g.*, **Fig. 1, 4**, and **Supplementary Fig. S2, S4, S5, S6**), which is therefore a plugged pre-pore rather than an open transmembrane pore.

On the same topic, the absence of vertical collapse during membrane insertion and pore formation, it is interesting to note that mobile membrane-attached mGSDMA3^{Nterm} oligomers (**Fig. R1**), membrane-inserted oligomers residing in the pre-pore and pore state all protrude by the same height above the membrane (**Fig. 1, 2, 4, new Supplementary Fig. S6**, and **revised Supplementary Table S1**). This suggests that neither the oligomer insertion nor the pore formation requires a vertical collapse. We have described this issue more clearly in our **revised Manuscript** (see revised Abstract, Results and Discussion).

Reviewer #2: Line 136: the height of GSDMA3 oligomers is 2.6 nm. The height of GSDMD is 3.6 nm (based on the Mulvihill 2018). How to explain this difference, since both proteins are structurally very similar?

Authors: Thank you very much for your observation. We apologize for the confusion created. We checked the possible reasons for this difference and found out that it is due to the different ways we measured the height of both gasdermin family members. The height of hGSDMD oligomers was measured manually by drawing a cross section across the highest protrusions of individual oligomer subunits and measuring the maximum height of the oligomer subunits above the membrane. The height of mGSDMA3, on the contrary, had been measured in an automated way by using a home-written MatLab tool. This tool averaged three arbitrarily taken cross sections through the center of the oligomer and not necessarily along the highest protrusions of the oligomeric subunits. When measuring the height of mGSDMA3 manually we obtain a value of 3.4 ± 0.3 nm (mean \pm SD, $n = 139$), which is closer to the height of hGSDMD oligomers inserted in *E. coli* polar lipid membranes (3.5 ± 0.3 nm)

reported in Mulvihill et al., 2018. To remove the confusion created in our initial submission, we have decided to manually remeasure all height values given in the **revised Manuscript**. The corrected height measures are now given in the main and supplementary figures and tables (**revised Fig. 1, 2, 4; new Supplementary Fig. S6 and revised Supplementary Table S1, S2**).

Reviewer #2: Line 140: Why are densities of membrane coverage different on images Fig1. A and B, and Fig. 2SA and B? Is this due to different concentration of GSDMA3 used? The concentrations of GSDMA3 in these panels should be specified in legends to corresponding figures, similar as for GSDMA3 and TEV, which is stated in the legend of Fig. S2 in description of panels C and D.

Authors: The concentrations of mGSDMA3 and TEV applied to SMLs are always the same but the binding is not homogeneous on the membrane. Naturally, while imaging we look for areas with higher binding density because they contain more information. The mGSDMA3 and TEV concentrations used are now specified in all figure legends and the Methods sections for AFM and TEM experiments (see **revised Figure Legends and Methods**).

Reviewer #2: Line 145: Legend to Fig. S3: the white arrowhead is mentioned (line 49), but I cannot see it on the image.

Authors: Thank you. We have removed the white arrow head from the legend of **Supplementary Fig. S3**. However, we have also **revised Supplementary Fig. S3** to show the oligomeric assembly at much improved detail.

Reviewer #2: Line 198: The assignment of prepores and pores may be tricky. Similar question as above for Fig. 1.

Authors: We thank the reviewer. The assignment of pre-pore and pore state was done by applying the same criteria as described for the analysis of **Fig. 1**. Briefly, knowing that the thickness of the EPL supported lipid membrane measures 5.0 ± 0.3 nm (mean \pm SD, $n=33$), we assume that oligomers having reduced the thickness of the membrane (lipids and sometimes also proteins) inside the pore by more than 2 nm have formed a transmembrane pore. The examples of mGSDMA3^{Nterm} oligomers shown in **Fig. 1, 4**, sometimes show the pore to be fully transmembrane (fully devoid of lipids or proteins). In other cases, lipids or proteins may still be inside the lumen of the oligomer (see high-resolution AFM topographs of the Manuscript, e.g. **Fig. 1, 4**, and **Supplementary Fig. S2, S4, S5, S6**), which is therefore a pre-pore rather than a functional transmembrane pore.

Reviewer #2: Line 216: Reference Ruan is not cited as other references. Please correct.

Authors: Thank you! The reference has now been cited properly.

Reviewer #2: Line 219: Fig. 4E shows comparison between AFM topograph and MD topograph for an oligomer with 30-fold stoichiometry. How oligomers with other different stoichiometries compare with the AFM topographs?

Authors: As outlined in **Supplementary Table S1**, we have conducted extensive MD simulations of various different oligomeric shapes and sizes. Among these, we repetitively simulated ring-shaped oligomers consisting of 18, 21, 27 and 30 mGSDMA3^{Nterm} for at least

1 μ s simulation time. In **Fig. R5** (included as **new Supplementary Fig. S10**) we compare the averages of AFM topographs and of MD simulations of ring-shaped oligomers consisting of 21(22), 27 and 30 mGSDMA3^{Nterm}. As we did not observe enough ring-shaped oligomers consisting of 21 mGSDMA3^{Nterm} (see **Fig. 4B**), we compared the averaged topograph of ring-shaped oligomers consisting of 22 mGSDMA3^{Nterm} imaged by AFM with MD simulations of a ring-shaped oligomer consisting of 21 mGSDMA3^{Nterm}. The comparisons show an excellent agreement between AFM topography and MD topography. However, the simulated ring-shaped oligomer consisting of 21 mGSDMA3^{Nterm} deformed slightly, which is in the line with our observation that too small oligomers destabilize and reshape (see also **Fig. 5A,B**).

Figure R5, included as new Supplementary Fig. S10 into the revised Manuscript. Comparison of topographs of ring-shaped oligomers encompassing 22/21, 27 or 30 mGSDMA3^{Nterm} as obtained from MD simulations and AFM. Top row, selected high-resolution AFM topographs (raw data) of individual ring-shaped mGSDMA3^{Nterm} oligomers inserted into SLMs and having stoichiometries of 22, 27 and 30. Middle row, symmetrized correlation averages of mGSDMA3^{Nterm} oligomers imaged by AFM and showing stoichiometries of 22, 27 and 30 (**Methods**). Bottom row, topographs of ring-shaped oligomers of 21, 27 and 30 mGSDMA3^{Nterm} each derived from averaging 1'800 oligomers from two MD simulations. MD simulations were averaged from snapshots taken every ns, after exclusion of the first 100 ns of the simulation. Raw data and averages of AFM topographs were taken from **Fig. 4**.

Reviewer #2: Line 267: Fig. S10- Line 149 in supp information: In the legend to Fig. S10, reference to Wimley is not cited as others. Please correct.

Authors: Thank you! The reference Wimley has now been cited properly.

Point-by-Point Response to Reviewer #3

Reviewer #3: Mari et al. apply various methods to study the pore-forming N-terminal domain of mouse gasdermin A3 (GSDMA3Nterm). AFM topographs reveal oligomers which self-assemble in arc-, slit-, and ring-shaped formations of pre-pores or transmembrane pores. Pre-pores are defined as oligomers with lipid-plugged apertures. The different structures formed immediately and remained stable over long time scales. These observations were confirmed for liposomes studied with TEM. Certain stoichiometries were more stable (18-36), with 30 being most common. Pore and pre-pore states existed indiscriminately for the three observed conformations, suggesting different stages in the pore-forming mechanism. To investigate further, Mari and colleagues used MD simulations. The simulations agree with the AFM data and further reveal that the tilt of the head domains relative to the β -hairpins optimize conditions, both chemical and structural, for a particular range of stoichiometries. The authors hypothesize that this introduces a size limit for exported biomolecules during lysis. Simulations also showed the process of pre-pore to lytic pore by flushing lipid from the opening by water. Lipids escape into the surrounding lipid, or into the solution as nanodiscs or liposomes.

Overall, the authors present a comprehensive study of the pore-formation mechanism of gasdermin-A3. The results will be of interest to a wide audience and showcase a powerful combination of techniques: AFM, TEM, and MD simulations. The experimental work, including especially the AFM, is of very high quality and the analysis appears solid. The work is well suited for eventual publication in Nat Communications.

Authors: We thank the reviewer for his/her supportive and constructive comments, which guided us to improve our manuscript. Below we answer point-by-point how we addressed each comment of the reviewer.

Reviewer #3: • Line 155: In what conditions were these samples stored for long-term experiments?

Authors: Thank you. Murine GSDMA3 (mGSDMA3) and TEV samples were aliquoted, flash-frozen and stored at -80°C . Aliquots were then used only once, just after allowing the sample to thaw on ice. EPL liposomes were also aliquoted, flash-frozen and stored at -80°C . We have now included the conditions to the revised Manuscript (see **revised Methods**, sections '*Cloning, Expression, and Purification of mGSDMA3 and TEV*' and '*Liposome Preparation*').

Reviewer #3: • Line 175: It should be stated how the number of subunits was determined for the oligomers. I'm assuming, since the data is very high resolution, that they could simply be counted the distinct gaussian-like peaks, either from the raw data or the correlation averages.

Authors: Thank you. The information has been added. Briefly, the AFM topographs of well-preserved ring-shaped oligomers were interpolated to become circular and their rotational power spectra allowed the number of subunits to be determined. Topographs with similar rotational power spectra were then correlation averaged. This procedure has been detailed in the **Methods** section '*AFM Image Analysis*'.

Reviewer #3: • Lines 218-9: What metric is used to assess the excellent agreement between the MD topograph and the AFM topograph?

Authors: We thank the reviewer and to answer the question we complement the text as follows: “Also, visual comparison revealed that the MD topograph of ring-shaped mGSDMA3^{Nterm} oligomers having 30-fold stoichiometry agreed excellently with the corresponding averaged AFM topograph at a resolution of ≈ 2 nm (Fig. 5E)”. Into our revised Manuscript we have also included a comparison between MD and AFM topographs of differently sized ring-shaped mGSDMA3^{Nterm} oligomers (new Supplementary Fig. S10).

Reviewer #3: • Lines 383-4: “... and 0.4 μ M TEV at 37 °C in imaging buffer (...) at RT.” It’s not clear here what temperature was used for this incubation – two temperatures are given.

Authors: Thank you. The information is now clearly given in our revised Manuscript. Briefly, we always cleave mGSDMA3 with TEV overnight at 37°C. Then there are 2 possibilities:

- 1) Incubating the membrane at 37°C for 3 h (with the mGSDMA3 solution previously digested overnight), then rinsing the sample thoroughly with buffer to remove non inserted protein from the solution, and imaging the sample by AFM at room temperature. Please note that room temperature is $\approx 33^\circ\text{C}$ since the AFM laser and the all instrumentation heat up the sample to this value.
- 2) Incubating the membrane at 37°C for 5–6 h (with the mGSDMA3 solution previously digested overnight), while imaging the membrane at 37°C. This is the case of the time-lapse AFM experiments, when we observe the mGSDMA3^{Nterm} assembling and inserting at physiological temperature, so the protein is still in solution while imaging and the temperature is 37°C. In the time-lapse AFM experiments the sample is kept and imaged at 37°C.

Reviewer #3: • Lines 428-9: How was drift determined here? Did you assume linear drift velocity and re-scale? Also, I guess that this drift varied day to day. It would be useful to know what was the range of drift rates observed in the instrument.

Authors: We thank the reviewer for this question. Usually, we acquire several high-resolution AFM topographs from the same area. By comparing these AFM topographs one can estimate the drift during the AFM imaging process. This estimation was applied to correct the drift of the individual images. We have revised the Methods section of our Manuscript to better explain how this drift correction was done (see revised Methods, section ‘AFM Image Analysis’).

Reviewer #3: • Line 580: “...the liposomes fused fused into membrane patches...” \Downarrow repeated word

Authors: Thank you, the repetitive wording has been removed.

Reviewer #3: • Line 586: “(M) The mixture...” \Downarrow panel label format: bold.

Authors: Thank you. The format has been changed into bold.

Reviewer #3: • Line 591: “(Methods)” \diamond is this a reference to the methods section?

Authors: Yes, it is a reference to the Methods section.

Reviewer #3: • There is no methods section for the TEM data. It is described in the Figure 2 caption, but not anywhere else.

Authors: We apologize for forgetting this section of the methods. We have now **revised the Methods** to include a section for the TEM, which reads:

Transmission Electron Microscopy (TEM)

For negative stain TEM, 5 μ l of sample was pipetted onto a glow-discharged copper grid coated with parlodion and carbon and left to adsorb for 1 min at RT. The grid was then washed with 4 droplets of nanopure water, and subsequently stained with 2 % uranyl acetate for 10 s, blotting between each step. Grids were scanned using a Tecnai G2 Spirit TEM microscope with a LaB6 filament operated at 120 kV (FEI Company, Eindhoven, The Netherlands). Images were recorded by a side-mounted EMSIS MORADA camera.

Reviewer #3: • General comment – two distinct modes of AFM imaging are discussed in the text and methods, but it is not clear which mode was used for each data set presented. To clarify, this information should be included in the captions.

Authors: Thank you. The information has been included into each figure legend (see **revised Figure Legends**).

	Connector loop 1	β -strand residue	Connector loop 2	Connector loop 3
GSDMD_MOUSE	GS	V	KEE	PAG
GSDMD_HUMAN	RS	V	QKE	PSG
GSDA3_MOUSE	GN	V	KQE	PKG
GSDMB_HUMAN	AE	I	VKE	PPN
GSDMA_HUMAN	GN	V	VQE	PKG
GSDME_HUMAN	G-	K	MQK	PAA
GSDME_MOUSE	S-	K	TQK	PAA
GSDMC_HUMAN	GP	F	INN	QKG
GSDMA_MOUSE	GN	V	LQE	PKG
GSDA2_MOUSE	GN	V	KQE	PKG
GSDME_HORSE	G-	K	TQK	PAP
GSDMC_MOUSE	AP	V	TKD	QKG
GSDC2_MOUSE	AP	V	SKD	PKG
GSDC3_MOUSE	AP	V	SKD	PKG
GSDC4_MOUSE	AP	V	SKD	PKG
GSDMC_RAT	AP	V	TKD	PKG

Table R1. Sequence alignment of the connector loops among 16 gasdermins.

Identity	mGSDMA3	hGSMA	mGSDMA	mGSDMA2	mGSDMD	hGSDMD
mGSDMA3	100%	74.03%	79%	78.23%	30.14%	32.92%
hGSDMA		100%	88.34%	75.78%	31.10%	32.71%
mGSDMA			100%	82%	30.72%	33.68%
mGSDMA2				100%	30.52%	31.82%
mGSDMD					100%	57.32%
hGSDMD						100%
Similarity	mGSDMA3	hGSMA	mGSDMA	mGSDMA2	mGSDMD	hGSDMD
mGSDMA3	100%	83.00%	84%	85.00%	49.00%	49.00%
hGSDMA		100%	94.00%	84.00%	50.00%	49.00%
mGSDMA			100%	87%	50.00%	50.00%
mGSDMA2				100%	48.00%	48.00%
mGSDMD					100%	72.00%
hGSDMD						100%

Table R2. Sequential identity and similarity among diverse members of the gasdermin D and gasdermin A families.

References

1. Engel, A., Schoenenberger, C.A. & Muller, D.J. High resolution imaging of native biological sample surfaces using scanning probe microscopy. *Curr Opin Struct Biol* **7**, 279-284 (1997).
2. Ruan, J., Xia, S., Liu, X., Lieberman, J. & Wu, H. Cryo-EM structure of the gasdermin A3 membrane pore. *Nature* **557**, 62-67 (2018).
3. Xia, S. et al. Gasdermin D pore structure reveals preferential release of mature interleukin-1. *Nature* **593**, 607-611 (2021).
4. Mulvihill, E. et al. Mechanism of membrane pore formation by human gasdermin-D. *EMBO J* **37**, e98321 (2018).
5. Ding, J. et al. Pore-forming activity and structural autoinhibition of the gasdermin family. *Nature* **535**, 111-116 (2016).
6. Sborgi, L. et al. GSDMD membrane pore formation constitutes the mechanism of pyroptotic cell death. *EMBO J* **35**, 1766-1778 (2016).
7. Dufrene, Y.F. et al. Imaging modes of atomic force microscopy for application in molecular and cell biology. *Nat Nanotechnol* **12**, 295-307 (2017).
8. Muller, D.J., Sass, H.J., Muller, S.A., Buldt, G. & Engel, A. Surface structures of native bacteriorhodopsin depend on the molecular packing arrangement in the membrane. *J Mol Biol* **285**, 1903-1909 (1999).
9. Muller, D.J. & Engel, A. Voltage and pH-induced channel closure of porin OmpF visualized by atomic force microscopy. *J Mol Biol* **285**, 1347-1351 (1999).
10. Engel, A. & Muller, D.J. Observing single biomolecules at work with the atomic force microscope. *Nat Struct Biol* **7**, 715-718 (2000).
11. Czajkowsky, D.M., Hotze, E.M., Shao, Z. & Tweten, R.K. Vertical collapse of a cytolysin prepore moves its transmembrane beta-hairpins to the membrane. *EMBO J* **23**, 3206-3215 (2004).
12. Palmer, M. et al. Assembly mechanism of the oligomeric streptolysin O pore: the early membrane lesion is lined by a free edge of the lipid membrane and is extended gradually during oligomerization. *EMBO J* **17**, 1598-1605 (1998).
13. Leung, C. et al. Stepwise visualization of membrane pore formation by suliyisin, a bacterial cholesterol-dependent cytolysin. *Elife* **3**, e04247 (2014).
14. Salvador-Gallego, R. et al. Bax assembly into rings and arcs in apoptotic mitochondria is linked to membrane pores. *EMBO J* **35**, 389-401 (2016).
15. Metkar, S.S. et al. Perforin oligomers form arcs in cellular membranes: a locus for intracellular delivery of granzymes. *Cell Death Differ* **22**, 74-85 (2015).
16. Mulvihill, E., van Pee, K., Mari, S.A., Muller, D.J. & Yildiz, O. Directly Observing the Lipid-Dependent Self-Assembly and Pore-Forming Mechanism of the Cytolytic Toxin Listeriolysin O. *Nano Lett* **15**, 6965-6973 (2015).
17. Sonnen, A.F., Pnitzko, J.M. & Gilbert, R.J. Incomplete pneumolysin oligomers form membrane pores. *Open Biol* **4**, 140044 (2014).
18. Vogele, M. et al. Membrane perforation by the pore-forming toxin pneumolysin. *Proc Natl Acad Sci U S A* **116**, 13352-13357 (2019).
19. Broz, P., Pelegrin, P. & Shao, F. The gasdermins, a protein family executing cell death and inflammation. *Nat Rev Immunol* **20**, 143-157 (2020).
20. Ding, J. & Shao, F. Growing a gasdermin pore in membranes of pyroptotic cells. *EMBO J* **37**, pii: embj.2018100067 (2018).
21. van Pee, K., Mulvihill, E., Muller, D.J. & Yildiz, O. Unraveling the Pore-Forming Steps of Pneumolysin from *Streptococcus pneumoniae*. *Nano Lett* **16**, 7915-7924 (2016).

Reviewers' Comments:

Reviewer #1:

Remarks to the Author:

The manuscript is much improved. They have undergone more systematic and thorough measurements as including in the revised figures and supplementary material. The authors have now addressed the issues of clarity I raised and have better put their work within the context of other published studies. I have no other queries.

Reviewer #2:

Remarks to the Author:

The authors have extensively modified the manuscript satisfactorily incorporated all the comments from reviewers. I have no further comments.

Reviewer #3:

Remarks to the Author:

I have no further comments or concerns about this manuscript.

NCOMMS-21-17423A-Z "Gasdermin-A3 pore formation propagates along variable pathways" Mari et al.

Point-by-Point Response to the Reviewer's Comments

Point-by-Point Response to Reviewer #1

Reviewer #1: The manuscript is much improved. They have undergone more systematic and thorough measurements as including in the revised figures and supplementary material. The authors have now addressed the issues of clarity I raised and have better put their work within the context of other published studies. I have no other queries.

Authors: We thank the reviewer for his/her critical and constructive comments, which have guided us to improve our manuscript.

Point-by-Point Response to Reviewer #2

Reviewer #2: The authors have extensively modified the manuscript satisfactorily incorporated all the comments from reviewers. I have no further comments.

Authors: We thank the reviewer for his/her critical and constructive comments, which have guided us to improve our manuscript.

Point-by-Point Response to Reviewer #3

Reviewer #3: I have no further comments or concerns about this manuscript.

Authors: We thank the reviewer for his/her critical and constructive comments, which have guided us to improve our manuscript.